# Internet-delivered transdiagnostic psychological treatments for individuals with depression, anxiety or both: a systematic review with meta-analysis of randomised controlled trials

Karoline Kolaas [1,2] Anne H Berman [1,3] Erik Hedman-Lagerlöf [4,5] Elin Lindsäter [4,5] Jonna Hybelius [2,6] Erland Axelsson [2,6]

For numbered affiliations see end of article.

**Correspondence to**
Dr Karoline Kolaas;
karoline.kolaas@ki.se

## ABSTRACT

**Objective** Depression and anxiety are major public health problems. This study evaluated the effects of internet-delivered transdiagnostic psychological treatments for individuals with depression, anxiety, or both.

**Design** Systematic review with meta-analysis.

**Data sources** Medline (Ovid), Cochrane Library (Wiley), the Web of Science Core Collection (Clarivate), and PsycInfo (EBSCO) were searched on 24 May 2021, with an update on 6 February 2023.

**Eligibility criteria** Randomised controlled trials of internet-delivered transdiagnostic psychological treatments, open to both participants with primary depression and participants with primary anxiety. This review concerned all treatment frameworks, both guided and unguided formats and all age groups.

**Data extraction and synthesis** In random-effects meta-analysis, we estimated pooled effects on depression symptoms and anxiety in terms of Hedges' $g$ with 95% CIs. Absolute and relative heterogeneity was quantified as the $\tau^2$ and $I^2$.

**Results** We included 57 trials with 21 795 participants. Nine trials (16%) recruited exclusively from routine care, and three (5%) delivered treatment via video. For adults, large within-group reductions were seen in depression ($g$=0.90; 95% CI 0.81 to 0.99) and anxiety ($g$=0.87; 95% CI 0.78 to 0.96). Compared with rudimentary passive controls, the added effects were moderate (depression: $g$=0.52; 95% CI 0.42 to 0.63; anxiety: $g$=0.45; 95% CI 0.34 to 0.56) and larger in trials that required all participants to meet full diagnostic criteria for depression or an anxiety disorder. Compared with attention/engagement controls, the added effects were small (depression: $g$=0.30; 95% CI 0.07 to 0.53; anxiety: $g$=0.21; 95% CI 0.01 to 0.42). Heterogeneity was substantial, and the certainty of the evidence was very low. Two trials concerned adolescents and reported mixed results. One trial concerned older adults and reported promising results.

**Conclusion** Internet-delivered transdiagnostic treatments for depression and anxiety show small-to-moderate added effects, varying by control condition. Research is needed regarding routine care, the video format, children and adolescents and older adults.

## STRENGTHS AND LIMITATIONS OF THIS STUDY

⇒ The systematic search strategy and the broad eligibility criteria allowed for a comprehensive overview of treatment frameworks, formats, age groups and settings including primary healthcare.

⇒ Several trial, participant and treatment characteristics were evaluated as potential moderators of the between-group effect versus rudimentary passive controls.

⇒ The majority of trials recruited self-referred participants through advertisement and were not conducted in clinical settings. This makes it more difficult to draw conclusions about effectiveness in routine care.

⇒ Our analyses of moderators were based exclusively on trial-level characteristics, and should preferably be replicated based on individual patient data.

**PROSPERO registration number** CRD42021243172.

## INTRODUCTION

Depression and the common anxiety disorders are associated with immense worldwide healthcare costs and disease burden.[1 2] Psychological treatments for depression and anxiety can be effective[3] including in clinical settings such as primary care,[4 5] where the majority of patients are found.[6] However, many patients with depression and anxiety disorders remain untreated or receive inadequate treatment,[7 8] illustrating the need for further development and dissemination of effective therapies.

It is widely recognised that depression and anxiety disorders often occur together,[9 10] and overlap in terms of genetic, cognitive and behavioural characteristics.[11–13] This speaks for the use of broader, transdiagnostic, psychological treatments designed to suit patients suffering from a range of different types of

depression and anxiety problems. That is, transdiagnostic treatments are characterised by focusing on components relevant for both depression and anxiety, in contrast to diagnosis-specific treatments targeting one specific mental health condition. In addition to beneficial effects for the patient, transdiagnostic treatments can also have organisational advantages, such as streamlined procedures and less time needed for diagnostic assessment.[14] In early meta-analysis, transdiagnostic psychological treatments have been found to have moderate pooled effects on depression and anxiety compared with mostly rudimentary treatment-as-usual and waitlist control groups.[15]

Another method of improving access to psychological treatments can be to deliver these online—for example via websites, mobile apps or videoconferencing—which saves time and costs for both clinicians and patients. For internet-delivered transdiagnostic psychological treatments, large within-group symptom reductions have been observed, with early meta-analyses reporting mixed outcomes compared with a heterogeneous range of control groups, and comparisons versus diagnosis-specific treatments typically result in relatively similar effects on depression and general anxiety.[16–18] Internet-delivered therapies appear to be accepted by patients, and have shown promising witin-group effects in pragmatic routine care studies.

Even though important advancements have been made in internet-delivered transdiagnostic psychological treatments for depression and anxiety in the past decades, several fundamental questions remain unanswered in this rapidly growing field. First, it remains unclear whether effects generalise to children, adolescents and older adults. Second, there has been no systematic comparison of treatment frameworks beyond the cognitive–behavioural therapies (CBT). Third, the relative effect of methods of delivery such as via websites, mobile apps and videoconferencing is yet unclear. Fourth, although acceptability and within-group change has been studied to some extent in the routine care environment,[19] based on randomised controlled trials (RCTs) that include a combination of patients with anxiety and depression, it is not known how well the effects of internet-delivered formats translate to routine clinical settings such as primary care.

We conducted a systematic review and meta-analysis of RCTs of internet-delivered transdiagnostic psychological treatments for depression and anxiety. This systematic review included all treatment frameworks, both guided and unguided formats and all age groups. Based on previous research, we hypothesised that between-group effects versus rudimentary passive controls would be large, and that the treatment length would be of little importance for the overall effect.

## METHOD
### Search strategy
This systematic review with meta-analysis adhered to the Preferred Reporting Items for Systematic reviews and Meta-Analyses (PRISMA) 2020 statement[20] and wasregistered on PROSPERO (CRD42021243172) on 16 April 2021 and at the Open Science Framework (https://osf.io/dtcey) on 18 May 2021, that is, before the initial database searches. We searched Medline (Ovid), Cochrane Library (Wiley), the Web of Science Core Collection (Clarivate) and PsycInfo (EBSCO) for RCTs of internet-delivered transdiagnostic psychological treatments for individuals with depression, anxiety or both. A search strategy was developed in collaboration with the Karolinska Institute library service. Candidate search strings were validated against a selection of articles that we were aware of prior to this systematic review, and after discussion within the research group, as well as with an external expert in psychodynamic online therapies. Four types of search terms were combined: terms for treatment (eg, 'psychotherapy'), terms for online communication (eg, 'internet-delivered'), terms for diagnostic focus (eg, 'transdiagnostic', 'depression' and 'anxiety') and terms for study design (eg, 'RCT'). Only publications published in 1995 or later were considered because there were no RCTs of internet-delivered transdiagnostic psychological treatments before then. Full search documentation for all databases can be found in online supplemental file A. An initial search was conducted on 24 May 2021, with an update on 6 February 2023 in accordance with Bramer and Bain.[21] We also reviewed the reference lists of previous meta-analyses that we found relevant.[16 17]

### Selection of studies
We employed a systematic deduplication algorithm in Endnote V.X9,[22] after which unique search hits were imported into Rayyan.[23] The publications were then assessed by two independent assessors in two phases. In phase I, all titles and abstracts were screened for eligibility by two authors (of KK, AHB, EH-L and EA), and non-relevant publications were excluded. In phase II, publications that had been included by at least one assessor in phase I were reviewed in full text. One assessor (KK) did this sequentially, that is, from criterion a to h (see below), for all full texts. Each full text was also reviewed by a second assessor (either AHB, EH-L, JH or EA) who suggested a decision of exclusion based on the first exclusion criterion identified, that is, in any order. Cases of disagreement were discussed in the research group and a final decision was reached either in consensus or, if necessary, by means of voting. If information about the eligibility criteria a–h was missing, authors were not contacted to provide additional information.

### Eligibility criteria
a. Studies had to be published in English in a peer-reviewed journal.
b. Studies had to be RCTs where the study participant was the unit of randomisation. This was because our focus was on conducting meta-analyses, for which cluster randomised trials require distinct analytical methods.[24]

Only one publication was formally included for each randomised trial.

c. Studies were required to evaluate the effect of an internet-delivered transdiagnostic psychological treatment for individuals with anxiety, depression or both. 'Internet-delivered' included treatments delivered via websites, mobile apps or videoconferencing. Treatments where online components were a mere complement to a face-to-face treatment, or where participants were required to attend a particular physical location, were excluded. 'Transdiagnostic' was defined as intending to target depression as well as anxiety. 'Psychological treatment' was defined as an intervention anchored in psychological theory with a clinical framing (1) where a clinician or facilitator had an active role in moderating the process, or (2) which was delivered in a structured manner, for example in the form of modules with suggested homework, or with an interactive component such as automatic feedback. For example, a webpage with general information about depression and anxiety would not be considered a psychological treatment, due to the lack of structure and interactivity.

d. Studies needed to recruit participants suffering from clinically significant depression, anxiety or both. We considered this criterion to be met if one of the following was true: (1) all participants were required to meet full diagnostic criteria for either primary depression or a primary anxiety disorder and both populations were included in the trial, or (2) all participants were required to score above a valid cut-off on an accepted screening measure for either depression or anxiety and both populations were included in the trial, (3) all participants were required to score above both a valid cut-off for depression and a valid cut-off for anxiety or (4) the baseline sample mean of at least one transdiagnostic treatment arm and at least one other arm were above both a valid cut-off for depression and a valid cut-off for anxiety. We excluded RCTs that either required all participants to meet full formal diagnostic criteria for depression, or required all participants to meet full formal diagnostic criteria for an anxiety disorder. This was to ensure a reasonably broad recruitment strategy, and to further reduce the risk that the treatment had focused overwhelmingly on one specific psychiatric disorder. We also excluded RCTs that required all participants to meet full formal diagnostic criteria for both depression and an anxiety disorder at the same time. This is a very unusual design that requires all participants to have a very substantial level of symptomatology that we suspected could be indicative of a highly specialised care setting unlike primary care which we were primarily interested in. Studies that subsumed obsessive–compulsive disorder or post-traumatic stress disorder under the umbrella of anxiety disorders, in accordance with earlier Diagnostic and Statistical Manual versions, were included. Studies that included a substantial proportion of subclinical

participants were included only if clinical participants were reported separately.

e. Studies were required to evaluate treatments for samples that were representative for a general medical or mental health clinical setting in the sense that the sample had not been heavily selected on any particular clinical characteristic or medical condition other than that of primary interest. This led to the exclusion, for example, of treatments specifically aimed at cancer patients with depression or anxiety.

f. Studies were required to have at least 10 participants in one arm enrolled in internet-delivered transdiagnostic psychological treatment. All control groups were included.

g. Studies were required to have employed at least one valid self-report questionnaire for depression or anxiety. If several valid questionnaires were reported, the one described as primary was used. If no questionnaire was described as primary, we tabulated that which was reported first in the manuscript. Domain-specific anxiety scales such as those concerning social anxiety, health anxiety or panic disorder anxiety were not considered adequate.

h. Post-treatment assessment needed to be completed within 3 months after treatment termination, so as to not lie closer to a typical follow-up assessment.

i. Studies were required to have reported means and SD, or provided information that made it possible to derive such estimates. Typically, this was based on the original publication. In certain cases, conversions were applied, for example when the SD could be derived from the sample size and the SE of the mean. Whenever necessary, the author was also contacted and encouraged to provide estimates via email.

## Data extraction

Prespecified variables were extracted by the first author (KK), including participant demographics (eg, age and gender). Studies were considered to concern children or adolescents if the mean age was below 18, and to concern older adults if the mean age was above 65. The following study and sample characteristics were extracted: study setting (primary care, not primary care or unclear), recruitment path (routine care or not routine care), main inclusion criterion (psychiatric diagnosis or 'cut-off symptoms'), mean depression severity (mild, moderate or severe) and mean anxiety severity (mild, moderate or severe). Symptom severity ratings were based on published norm data.[25–28] Extracted treatment characteristics were as follows: delivery format (website, mobile app, video, mixed or other), therapist support (guided or unguided), treatment type (CBT, mindfulness, mixed CBT and mindfulness or other), specific treatment protocol, treatment length (≥6 weeks or <6 weeks; intended to contrast what are usually considered brief therapies with therapies of a more conventional length), component flexibility (standardised or tailored) and therapist time (minutes per participant in treatment). In line

with recent findings demonstrating that many treatment-as-usual control groups are virtually indistinguishable from waitlist controls,[29] we did not lump all treatment-as-usual controls into the same category but rather made a distinction among (a) rudimentary passive controls that comprised waitlist controls and other controls without a structured intervention, (b) attention/engagement controls where the participants received some sort of standardised intervention that controlled for the attention from a caregiver or engagement in a study but without this being a conventional psychological treatment and (c) bona-fide treatments 'delivered by trained therapists and […] based on psychological principles, […] offered to the psychotherapy community as viable treatments'.[30] For each trial, the primary outcome for depression and anxiety was tabulated. If no outcome was defined as primary, or the primary outcome measure was deemed not to be valid, we tabulated the first valid measure that was mentioned. KK contacted the corresponding author when main outcome data were missing. All authors were also asked to provide pre–post correlations for the calculation of within-group effects. Intention-to-treat estimates were tabulated whenever available.

### Assessment of risk of bias and quality of the evidence base

Trial risk of bias was assessed using a modified version of the Cochrane risk-of-bias tool for randomised trials V.2 (RoB 2).[31] The RoB 2 covers five domains: (1) the randomisation process, (2) deviations from intended interventions, (3) missing outcome data, (4) measurement of the outcome and (5) selection of the reported results. Because of the difficulty in blinding therapists to the psychological treatment they are delivering, we omitted items related to blinding. Consistent with previous similar research, non-blinding did not therefore lead to an elevated risk of bias.[15 32] Each trial received one rating on each risk of bias domain, and this rating concerned both the depression and the anxiety outcome. Reporting one score was sufficient because all trials administered the depression and anxiety measures in parallel, which resulted in identical ratings. Authors KK and EL independently assessed the trials and discussed cases of disagreement to reach consensus. The decision was made that if five articles in a row were rated identically by both assessors, it would no longer be necessary with two assessments per article. This target was achieved after 25 articles, and KK assessed the remaining trials. Based partially on the risk of bias ratings, we assessed the overall quality of the evidence base according to Grading of Recommendations Assessment, Development and Evaluation (GRADE).[33] We assigned separate ratings for the evidence base pertaining to (1) effects versus rudimentary passive controls for adults, (2) effects versus attention/engagement controls for adults, (3) effects versus other bona-fide treatments for adults, (4) effects for children and adolescents and (5) effects in primary care. Similar to the risk of bias assessment, one common score was given for depression and anxiety.

### Statistical analysis

We report studies focusing on children and adolescents, and studies focusing on older adults, separately due to their specific focus. Effects versus other bona-fide treatments were too few to be included in meta-analysis. At the end of the tabulation phase, we also decided to report studies with a treatment length of 2 weeks or shorter separately, a duration which would not typically be regarded as bona fide in routine clinical care. Nine randomised factorial trials were also included in the meta-analysis of within-group effects, but their main results were summarised separately.

We proceeded with meta-analysis using random-effects models fitted with the restricted maximum likelihood estimator using the inverse variance method, in R V.4.2.0[34] with the metafor 3.8–1 package.[35] Effects were quantified in terms of Hedges' $g$, that is, the bias-corrected standardised mean difference.[36] Although not our primary focus, we first conducted a meta-analysis of the within-group change in treatment. The point of this was to ensure that change had been observed per se, to determine if heterogeneity was much reduced without the influence of variance due to different control groups, and to facilitate the interpretation of within-group effects in future studies. For this analysis, we standardised $g$ on the pretreatment SD, and used the pre–post correlation (imputed pooled estimate if missing) to determine the sampling variance:[37 38]

$$g = \frac{M_{pre} - M_{post}}{SD_{pre}} \times \sqrt{\frac{2}{(n-1)}} \left( \frac{r[(n-1)/2]}{r[(n-2)/2]} \right)$$

$$approx.variance(g) = \frac{2(1 - r_{prepost})}{n} + \frac{g^2}{2n}$$

We then conducted a meta-analysis of between-group effects versus rudimentary passive controls, and a meta-analysis of between-group effects versus attention/engagement controls. For these analyses, we standardised $g$ on on the pooled post-treatment SD, and employed the large-sample approximation of the sampling variance:[36 39]

$$g = -\frac{M_{tx} - M_{contol}}{SD_{pooled}} \times \sqrt{\frac{2}{(n_{tx} + n_{contol} - 2)}} \left( \frac{r[(n_{tx} + n_{control} - 2)/2]}{r[(n_{tx} + n_{control} - 3)/2]} \right)$$

$$approx.variance(g) = \frac{n_{tx} + n_{control}}{n_{tx} n_{control}} + \frac{g^2}{2(n_{tx} + n_{control})}$$

Absolute values for $g$ of 0.2 are usually regarded as small, 0.5 as moderate and 0.8 as large.[40] We quantified absolute heterogeneity in terms of the $\tau^2$, and also report the $I^2$ which stands for the proportion of the between-sample variance that is due to true study differences as opposed to sampling error. An $I^2$ of 25% is usually regarded as low, 50% as moderate and 75% as high,[41] though the $I^2$ also increases with the precision of the original trials.[42] Pertaining to the meta-analysis of trials of transdiagnostic treatment versus rudimentary passive controls, a series of planned moderator analyses were conducted on the basis of trial, sample and treatment characteristics whenever strata included at least four studies. One exception was made to this rule where we analysed primary care setting as a potential moderator based on four arms from three

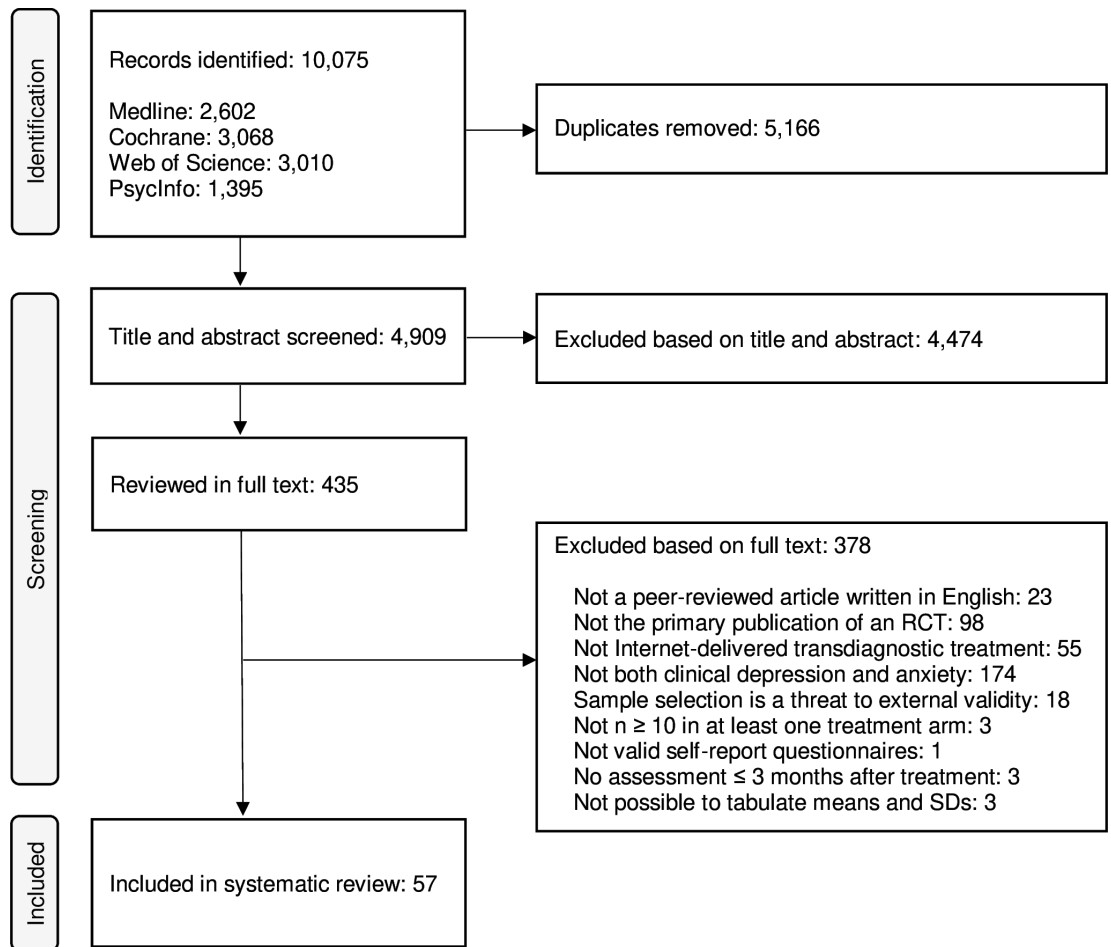

**Figure 1** PRISMA flowchart of the study selection process. PRISMA, Preferred Reporting Items for Systematic reviews and Meta-Analyses; RCT, randomised controlled trial.

trials because we found this test important. In the analysis of trials with rudimentary controls, we also investigated publication bias based on the visual inspection of funnel plots in combination with Egger's test[43] and the Duval and Tweedie trim and fill procedure.[44] In line with recommendations,[45] the latter was repeated using the R0, L0 and Q0 estimators.

### Patient and public involvement
None

### RESULTS
The inclusion process is presented in figure 1. Out of 435 RCTs reviewed in full text, 55 were suggested for inclusion by both assessors, 335 were immediate exclusions and 45 were disagreements (90% agreement; κ=0.65). The reasons for these disagreements are listed in online supplemental file B. Of the disagreements, five trials were included after discussion. Three trials were then excluded due to insufficient data. This resulted in a final sample of 57 RCTs. Full references to these trials are found in online supplemental file 1.

### Study characteristics
In the 57 included trials, 21 795 participants were randomised.[46–102] Arms were as follows: 83 internet-delivered transdiagnostic psychological treatments, 36 rudimentary controls, 8 attention/engagement controls and three other bona-fide treatments. With regard to age groups, 53/57 trials (93%) recruited adults, 3/57 (5%) children or adolescents and 1/57 (2%) specifically older adults. As to delivery formats, 37/57 (65%) of the internet-delivered transdiagnostic psychological treatments were delivered via a website mainly through text, 8/57 (14%) via a mobile app, 3/57 (5%) via video and 9/57 (16%) were mixed or other. In terms of the therapeutic model, 37/57 (65%) evaluated some variation on CBT, 8/57 (14%) mindfulness-based therapy, 3/57 (5%) mixed CBT and mindfulness and 11/57 (19%) other frameworks such as problem-solving therapy or affect-focused psychodynamic therapy. As for the recruitment strategy, 39/57 (68%) trials relied on self-referrals, 9/57 (16%) recruited from routine care including student counselling and 9/57 (16%) used a mixed strategy or other methods such as recruitment via a public health-care website. Illustrating the lack of work in the routine clinical setting, 54/57 trials (95%) did not report having

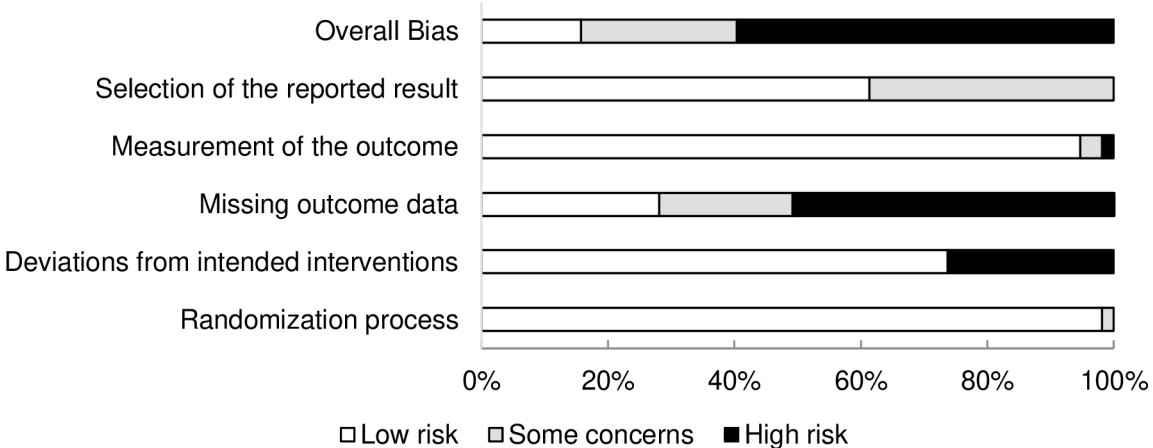

**Figure 2** Study risk of bias based on the Cochrane risk-of-bias tool version 2. Note that blinding was not assessed, which means that the lack of blinding did not result in an elevated risk score.

been conducted in primary healthcare and 3/57 (5%) reported having been conducted in primary healthcare. The trials were based in Australia 15/57 (26%), Canada 12/57 (21%), the USA 6/57 (11%,), UK 5/57 (9%), Sweden 4/57 (7%), the Netherlands 3/57 (5%) and other countries 12/57 (21%). The most common self-report questionnaire measuring depression symptoms was the Patient Health Questionnaire 9 (PHQ-9) 31/57 (54%), followed by the Depression, Anxiety and Depression Scale 21 (DASS-21) 11/57 (19%). The most common anxiety measure was the GAD-7 32/57 (56%), followed by the DASS-21 11/57 (19%) and the Hospital Anxiety and Depression Scale-Anxiety subscale (HADS-A) 6/57 (11%). More trial characteristics are tabulated in the online supplemental file B.

### Study risk of bias and quality of the evidence base

The distribution of study risk-of-bias ratings is presented in figure 2. Overall, 9/57 (16%) of the trials were rated as having a low risk of bias, 14/57 (25%) as raising some concerns and 34/57 (60%) as having a high risk of bias. The most common reason for an overall high risk was missing outcome data, which was seen in 29/59 (51%) of the trials, followed by deviations from the intended interventions in 15/57 (26%). Pooled effect sizes did not differ between trials as a function of the overall risk of bias (see table 1). In all outcome domains, the overall quality of the evidence base was very low (see online supplemental file C) for details).

### Effects on depression and anxiety

For adults, the pooled within-group reduction in depression symptoms was large ($g$=0.90; 95% CI 0.81 to 0.99; $k$=78) and the heterogeneity was high ($I^2$=95%; $\tau^2$=0.14; $Q_{77}$=1606, $p \leq 0.001$). The pooled within-group reduction in anxiety was also large ($g$=0.87; 95% CI 0.78 to 0.96; $k$=78) and the heterogeneity was high ($I^2$=95%; $\tau^2$=0.16; $Q_{77}$=1882, $p \leq 0.001$). The pooled between-group effect versus the rudimentary controls was moderate for depression symptoms ($g$=0.52; 95% CI 0.42 to 0.63; $k$=43) with high heterogeneity ($I^2$=80%; $\tau^2$=0.08; $Q_{42}$=162, $p$

$\leq 0.001$), and moderate for anxiety ($g$=0.45; 95% CI 0.34 to 0.56; $k$=43) with high heterogeneity ($I^2$=83%; $\tau^2$=0.10; $Q_{42}$=176, $p \leq 0.001$). See figure 3. The pooled between-group effect versus the attention/engagement controls was also significant but small for depression symptoms ($g$=0.30; 95% CI 0.07 to 0.53; $k$=10; $I^2$=91%; $\tau^2$=0.12; $Q_9$=48, $p \leq 0.001$) and small for anxiety ($g$=0.21; 95% CI 0.01 to 0.42; $k$=10; $I^2$=88%; $\tau^2$=0.09; $Q_9$=45, $p \leq 0.001$). See figure 4.

### Risk of publication bias

We explored indicators of publication bias in the meta-analysis of effects versus rudimentary passive controls. For depression, there was clear indication of small-to-medium sized trials with non-beneficial effects of internet-delivered psychological treatments being under-reported, and with imputed studies the between-group effect dropped from 0.52 to 0.29 (95% CI 0.16 to 0.42). For anxiety, there were also indication of possible asymmetry in the same direction, but this was more ambiguous, and the trim-and-fill procedure did not result in any imputed trials. See online supplemental file B for details.

### Moderators and subgroups

The results of the planned moderator and subgroup analyses are presented in table 1. With regard to between-group effects versus the rudimentary controls, one moderator was found to be significant for both depression and anxiety symptoms: trials that required all participants to meet full criteria for a psychiatric diagnosis reported a larger pooled between-group effect as compared with trials that did not have this requirement. We also identified nine randomised factorial trials. These manipulated various aspects of the treatment format reported mostly null results with small effect sizes. In single trials, small moderating effects were reported for therapist guidance, app recommendations and pretreatment motivational interviewing.[70 96] In other trials, null results were, however, also reported for guidance and motivational interviewing.[51 80]

**Table 1** Moderator and subgroup analyses of the relative effect of internet-delivered transdiagnostic psychological treatment versus rudimentary controls

### Depression symptoms

| Potential moderator | Q (df) | P value | Subgroup | k | g | 95% CI | $I^2$ (%) | $\tau^2$ |
|---|---|---|---|---|---|---|---|---|
| Pooled total | | | | 43 | 0.52 | 0.42 to 0.63 | 80 | 0.08 |
| *Study and sample characteristics* | | | | | | | | |
| Study setting | 2.14 (1) | 0.144 | Primary healthcare | 4 | 0.30† | −0.03 to 0.62 | 79 | 0.08 |
| | | | Not or unclear | 39 | 0.55 | 0.44 to 0.66 | | |
| Recruitment path | 3.32 (1) | 0.068 | Routine care | 8 | 0.32 | 0.07 to 0.57 | 81 | 0.09 |
| | | | Not routine care | 31 | 0.58 | 0.45 to 0.71 | | |
| Main inclusion criterion | 6.33 (1) | 0.012* | Psychiatric diagnosis | 12 | 0.74 | 0.54 to 0.94 | 78 | 0.08 |
| | | | Cut-off or symptoms | 31 | 0.44 | 0.32 to 0.56 | | |
| Depression severity | 2.31 (2) | 0.315 | Mild | 5 | 0.46 | 0.18 to 0.74 | 68 | 0.06 |
| | | | Moderate | 32 | 0.53 | 0.42 to 0.64 | | |
| | | | Severe | 4 | 0.28† | −0.04 to 0.59 | | |
| Anxiety severity | 1.29 (1) | 0.255 | Mild or moderate | 30 | 0.46 | 0.35 to 0.56 | 74 | 0.06 |
| | | | Severe | 10 | 0.59 | 0.38 to 0.79 | | |
| *Treatment characteristics* | | | | | | | | |
| Delivery format | 1.51 (2) | 0.471 | Website | 29 | 0.50 | 0.38 to 0.63 | 79 | 0.08 |
| | | | Mobile app | 6 | 0.43 | 0.09 to 0.77 | | |
| | | | Mixed or other | 8 | 0.66 | 0.41 to 0.90 | | |
| Therapist support | 0.91 (1) | 0.341 | Yes, guided | 23 | 0.58 | 0.43 to 0.73 | 80 | 0.08 |
| | | | No, unguided | 20 | 0.47 | 0.32 to 0.62 | | |
| Treatment type | 4.67 (2) | 0.097 | CBT | 29 | 0.53 | 0.41 to 0.66 | 77 | 0.27 |
| | | | Mindfulness | 4 | 0.88 | 0.49 to 1.27 | | |
| | | | Other | 10 | 0.40 | 0.21 to 0.60 | | |
| Specific protocols | 0.06 (1) | 0.812 | Well-being course | 6 | 0.49 | 0.21 to 0.78 | 81 | 0.09 |
| | | | Not well-being course | 37 | 0.53 | 0.41 to 0.65 | | |
| Treatment length | 0.02 (1) | 0.901 | ≥6 weeks | 28 | 0.53 | 0.40 to 0.66 | 79 | 0.09 |
| | | | <6 weeks | 15 | 0.52 | 0.33 to 0.70 | | |
| Component flexibility | 1.00 (1) | 0.318 | Standardised | 36 | 0.56 | 0.44 to 0.68 | 81 | 0.09 |
| | | | Tailored | 5 | 0.39 | 0.08 to 0.70 | | |
| Therapist time‡ | 0.88 (1) | 0.349 | Per additional 10 min | 28 | 0.00† | −0.01 to 0.01 | 63 | 0.03 |
| *Overall study risk of bias* | | | | | | | | |
| Cochrane tool version 2 | 0.84 (2) | 0.658 | High | 29 | 0.49 | 0.36 to 0.63 | 81 | 0.09 |
| | | | Some concerns | 8 | 0.55 | 0.30 to 0.80 | | |
| | | | Low | 6 | 0.63 | 0.35 to 0.91 | | |

### Anxiety

| Potential moderator | Q (df) | P value | Subgroup | k | g | 95% CI | $I^2$ | $\tau^2$ |
|---|---|---|---|---|---|---|---|---|
| Pooled total | | | | 43 | 0.45 | 0.34 to 0.56 | 83 | 0.10 |
| *Study and sample characteristics* | | | | | | | | |
| Study setting | 0.91 (1) | 0.340 | Primary healthcare | 4 | 0.28† | −0.07 to 0.64 | 83 | 0.10 |
| | | | Not or unclear | 39 | 0.47 | 0.35 to 0.59 | | |
| Recruitment path | 2.19 (1) | 0.139 | Routine care | 8 | 0.30 | 0.04 to 0.55 | 82 | 0.10 |
| | | | Not routine care | 31 | 0.52 | 0.38 to 0.65 | | |
| Main inclusion criterion | 7.01 (1) | 0.008* | Psychiatric diagnosis | 12 | 0.68 | 0.48 to 0.89 | 81 | 0.09 |
| | | | Cut-off or symptoms | 31 | 0.36 | 0.23 to 0.48 | | |
| Depression severity | 2.60 (2) | 0.272 | Mild | 5 | 0.51 | 0.20 to 0.81 | 74 | 0.08 |

Continued

**Table 1** Continued

| Anxiety | | | | | | | | |
|---|---|---|---|---|---|---|---|---|
| | | | Moderate | 32 | 0.44 | 0.32 to 0.56 | | |
| | | | Severe | 4 | 0.16† | −0.18 to 0.50 | | |
| Anxiety severity | 0.85 (1) | 0.358 | Mild or moderate | 30 | 0.38 | 0.26 to 0.50 | 79 | 0.07 |
| | | | Severe | 10 | 0.50 | 0.28 to 0.72 | | |
| *Treatment characteristics* | | | | | | | | |
| Delivery format | 0.02 (2) | 0.988 | Website | 29 | 0.45 | 0.31 to 0.59 | 83 | 0.11 |
| | | | Mobile app | 6 | 0.42 | 0.06 to 0.79 | | |
| | | | Mixed or other | 8 | 0.46 | 0.19 to 0.73 | | |
| Therapist support | 1.67 (1) | 0.196 | Yes, guided | 23 | 0.52 | 0.36 to 0.68 | 82 | 0.10 |
| | | | No, unguided | 20 | 0.38 | 0.22 to 0.53 | | |
| Treatment type | 5.57 (2) | 0.062 | CBT | 29 | 0.44 | 0.31 to 0.57 | 80 | 0.29 |
| | | | Mindfulness | 4 | 0.89 | 0.48 to 1.30 | | |
| | | | Other | 10 | 0.34 | 0.13 to 0.55 | | |
| Specific protocols | 0.96 (1) | 0.328 | Well-being course | 6 | 0.31 | 0.00 to 0.61 | 84 | 0.10 |
| | | | Not well-being course | 37 | 0.47 | 0.35 to 0.60 | | |
| Treatment length | 0.00 (1) | 0.973 | ≥6 weeks | 28 | 0.45 | 0.31 to 0.59 | 83 | 0.11 |
| | | | <6 weeks | 15 | 0.45 | 0.25 to 0.66 | | |
| Component flexibility | 2.22 (1) | 0.136 | Standardised | 36 | 0.49 | 0.36 to 0.61 | 84 | 0.11 |
| | | | Tailored | 5 | 0.22† | −0.11 to 0.55 | | |
| Therapist time‡ | 0.47 (1) | 0.494 | Per additional 10 min | 28 | 0.00† | −0.01 to 0.02 | 85 | 0.09 |
| *Overall study risk of bias* | | | | | | | | |
| Cochrane tool version 2 | 0.01 (2) | 0.995 | High | 29 | 0.45 | 0.31 to 0.60 | 84 | 0.11 |
| | | | Some concerns | 8 | 0.44 | 0.17 to 0.71 | | |
| | | | Low | 6 | 0.46 | 0.16 to 0.76 | | |

In accordance with the study protocol, strata were collapsed or excluded from analysis if $k < 4$. For anxiety severity, 'mild' ($k = 3$) and 'moderate' were collapsed. For delivery formats, 'video' ($k = 3$) was subsumed under 'mixed or other'. For treatment types, 'CBT and mindfulness' ($k = 3$) was subsumed under 'other'. Q-tests listed in this table are tests of moderation, with the intercept included in the statistical model. Heterogeneity is reported as $I^2$ and $\tau^2$. CBT, cognitive behavior therapy.
*Pooled effect not significantly different from zero.
†Test of moderation significant at $\alpha = 0.05$.
‡Therapist time per participant was tested as a continuous variable.
CBT, cognitive–behavioural therapies.

### Effects versus other bona-fide treatments

Three trials compared the effects of an internet-delivered transdiagnostic psychological treatment with those of a distinctly different bona-fide treatment. These trials showed mixed results and are described in online supplemental file B.[49 79 91]

### Children, adolescents and older adults

Two trials recruited children or adolescents. One trial recruited ages 12–21 (M=16, SD=2; n=45) and reported null effects of 5 weeks of problem-solving therapy on depression ($g=0.04$) and anxiety ($g=−0.11$) as compared with a waitlist.[62] The second trial recruited ages 13–18 (M=16, SD=1; n=103) in a school setting and found that a single session promoting growth, gratitude and values had significant and moderate added effects on depression ($g=0.50$), but not anxiety ($g=0.32$) as compared with a control that focused on the promotion of study skills.[74] One trial exclusively recruited older adults, aged 60+ (M=66, SD=5; n=433) and found that 8 weeks of CBT in a structured text-based self-help format was associated with large within-group reductions in depression ($g$s=1.19–1.30) and anxiety ($g$s=1.21–1.31), but that there was no significant added effect of adding guidance or an initial interview.[85]

### DISCUSSION

In this systematic review with meta-analysis based on 57 RCTs we found that, for adults, internet-delivered transdiagnostic psychological treatments led to moderate between-group effects compared with rudimentary controls (depression: $g=0.52$; anxiety: $g=0.45$) and small between-group effects compared with attention/

### Standardized effect on depression symptoms

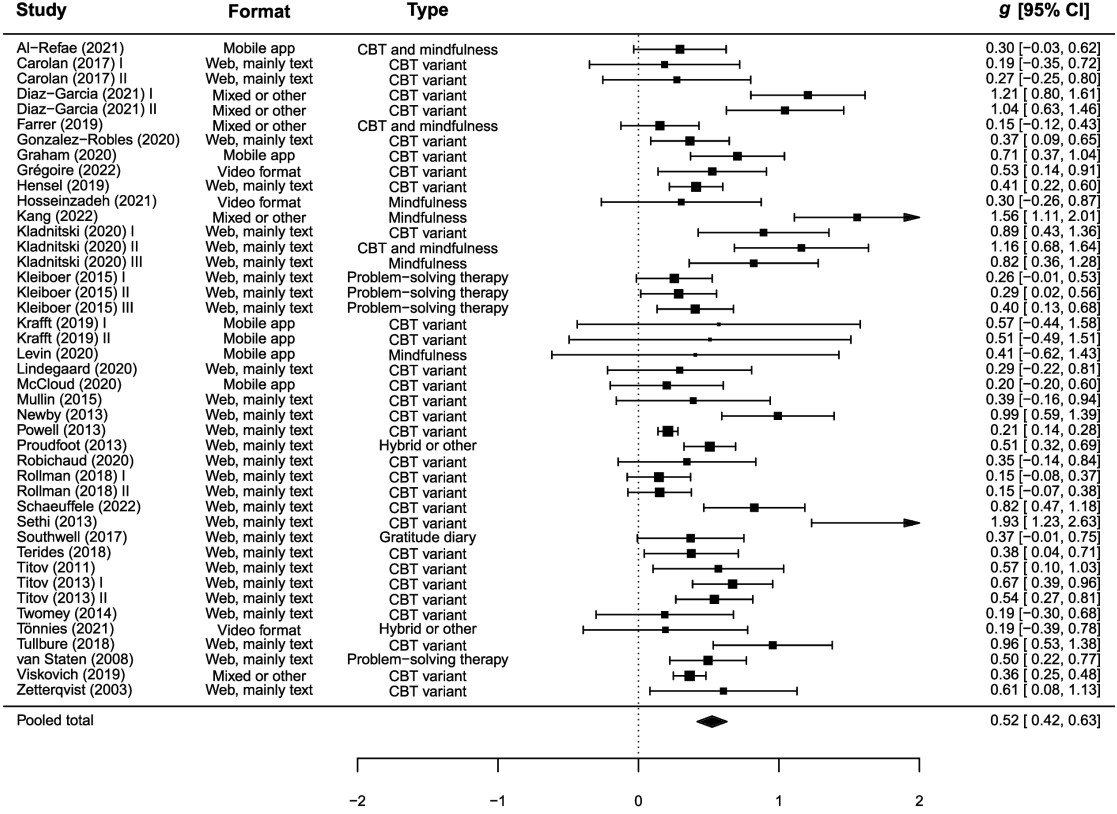

Higher values favour internet-delivered psychological treatment vs. rudimentary passive controls

### Standardized effect on anxiety

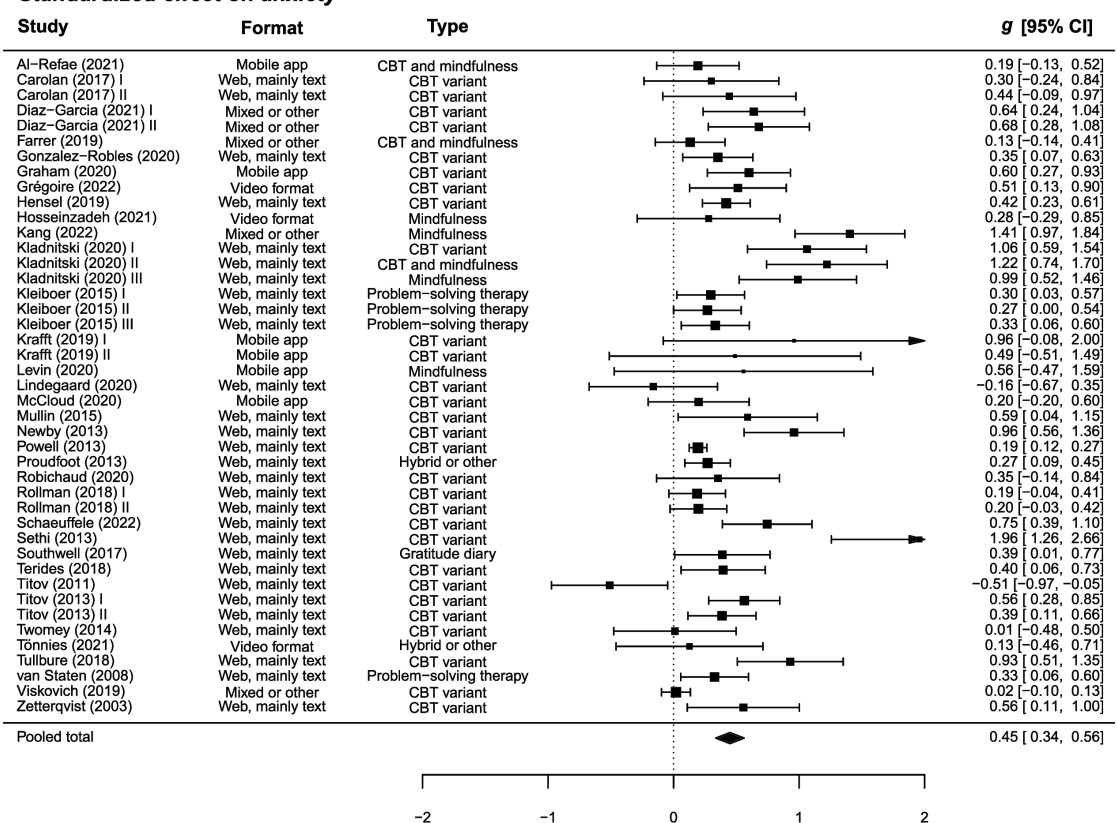

Higher values favour internet-delivered psychological treatment vs. rudimentary passive controls

**Figure 3**  Standardised effect on depression and anxiety symptoms versus rudimentary passive controls.

### Standardized effect on depression symptoms

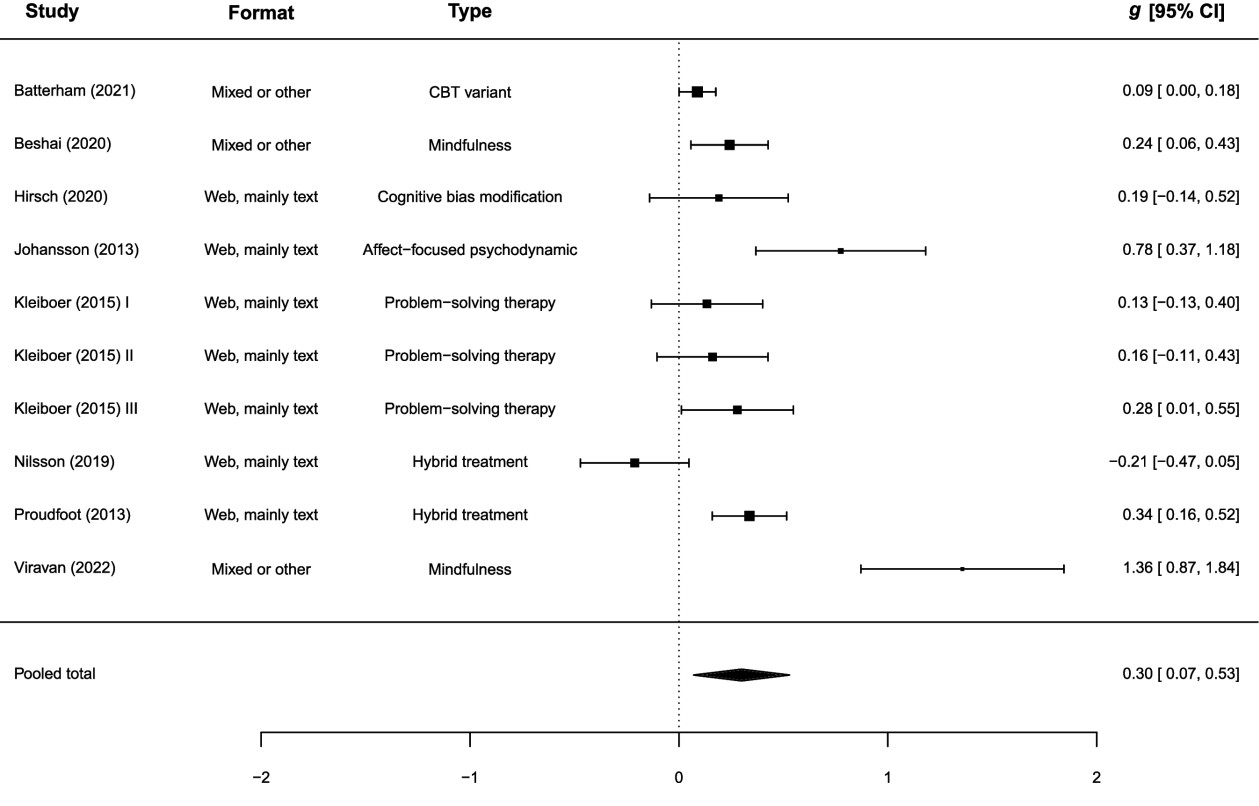

### Standardized effect on anxiety

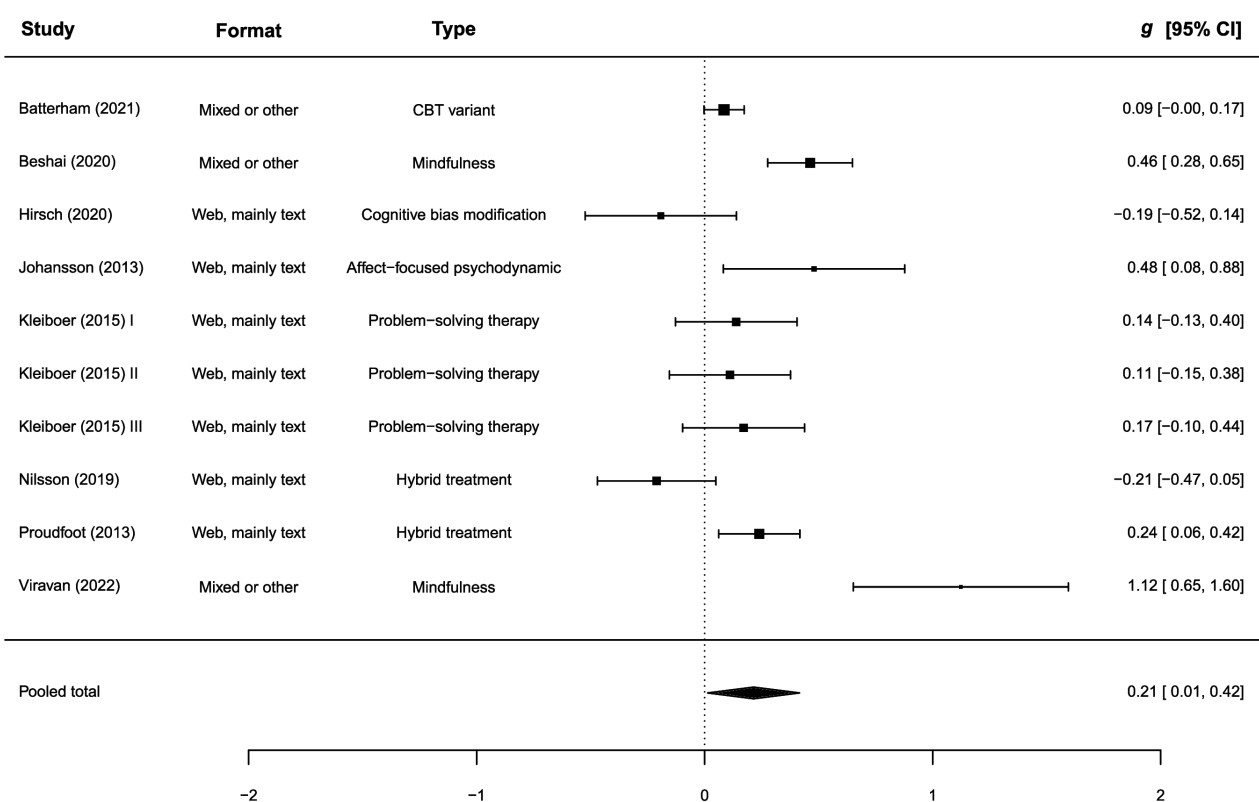

**Figure 4**  Standardised effect on depression and anxiety symptoms versus attention/engagement controls.

engagement controls (depression: $g=0.30$; anxiety: $g=0.21$). Heterogeneity was substantial. There were also indications of possible publication bias, which probably inflated the pooled effect on depression versus rudimentary controls—perhaps as much as 0.23 units. One significant moderator was found: trials where all participants were required to meet full diagnostic criteria for depression or an anxiety disorder reported larger between-group effects versus rudimentary controls, than did trials without this requirement. Strengths of the present study include the systematic search strategy and the broad eligibility criteria, which allowed for an overview of treatment frameworks, formats, settings including primary healthcare and also age groups. On the whole, the overall quality of the evidence base was very low, which implies that there is considerable uncertainty about true effects.

### Comparison with prior research
Our final sample of 57 RCTs pertaining specifically to internet-delivered transdiagnostic psychological treatments suitable for both primary depression and primary anxiety disorders is large compared with previous systematic reviews.[15] This is probably explained both by the rapid growth of this research field, and our use of exhaustive search terms developed with the specific aim of identifying internet-delivered transdiagnostic psychological treatments. The between-group effect versus rudimentary controls was slightly smaller than expected, though still broadly in line with previous meta-analyses.[15 16] Our expectation, based on previous research,[17] that treatment length would be of little importance for the between-group effect, was also supported in our data. Similarly, the finding that effects on depression are likely to be inflated by publication bias is also in line with previous research and therefore not unexpected.[103]

### Children, adolescents and older adults
This exhaustive literature search showed very clearly that the internet-delivered transdiagnostic treatments for children and adolescents, as well as specifically for older adults, have not come very far. Especially for children and adolescents, we found this rather surprising considering the relatively long tradition of treatments intended to suit a wide spectrum of anxiety, but not necessarily mood, disorders.[104] Conceivably, this may have reduced the incentive to develop newer protocols that can accommodate patients with depression as well. For older adults, we identified a single study which presented promising results. It is widely acknowledged that older adults rarely receive psychological treatment.[105 106] We wish to encourage further work in this area, especially with increased technical prowess in this population.

### Possible benefit of meeting a diagnosis
In the exploratory moderator analyses, the only variable that was a significant moderator of the between-group effect of treatment was whether participants were required to meet full criteria for a psychiatric diagnosis.

One interpretation is that of a 'specificity effect', where larger effects can be expected if patients suffer from clinical depression or an anxiety disorder, as opposed to symptoms caused for example by a somatoform disorder or a somatic disease.

### Treatment characteristics
Except for the possible benefits of a diagnosis, a general pattern in both the planned tests of potential moderators and the original factorial trials was the lack of significant moderators, not least in terms of treatment characteristics. One interesting finding was for example that therapist-guidance was not found to be a significant moderator in our meta-analyses, and that results from the factorial trials were inconsistent. We found that it is still unusual for trials to investigate the effect of treatments other than CBT or mindfulness-based interventions. There were, however, also examples of other treatments investigated in more than one trial, namely problem-solving therapies[62 65 89] and bias-modification training therapies.[61 97]

### New formats: are we at a crossroads?
For the past quarter of a century, internet-delivered psychological treatments have mostly been delivered via web platforms, with standardised linear text presentations intended to be accessed via a web browser. This systematic review illustrates that this treatment format is still most widely researched, and that newer formats remain understudied. We were able to pool the mean effects of six arms from five trials focusing on mobile apps, and five arms from five trials of tailored treatments, which showed no significant differences compared with the more common standardised web format. Notably, only three trials investigated videoconferencing interventions,[86 93 100] and only one trial investigated the effect of an intervention using artificial intelligence.[54] The vast majority of the literature still pertains to the standardised web-based text format, but the direction of the field appears to be towards automated, tailored, interventions.

### Limitations
A clear limitation of the original studies was that the majority of trials recruited self-referred participants through advertisement and were not conducted in clinical setting. The fact that these samples might have been highly motivated, and not representative of patients in routine care, makes it more difficult to draw conclusions about effectiveness in routine care. Another limitation of this review is that our analyses of moderators were based exclusively on trial-level characteristics. This implies poor power, and often restriction of range, in many tests such as that of whether pooled effects differ as a function of mean baseline symptom level. This review included studies published in English only, which means that RCTs published solely in other languages were systematically overlooked. Another aspect that warranted lowered GRADE ratings was the considerable heterogeneity in reported effects, which remained largely unexplained

and often implied poor precision. Last, we remind the reader that being enrolled in a psychological treamtent implies engaging with the content of that treatment, which often includes its theoretical basis. Therefore, therapists and patients cannot be fully blinded to the intervention, and effects may depend partially on expectancy effects, including therapist's and patient's presumptions about 'what ought to work'.

## Clinical implications and recommendations for future research

This meta-analysis contributes tentative evidence that transdiagnostic psychological treatments are effective for people with elevated depression symptoms or anxiety. However, even though a substantial proportion of patients are treated in routine care settings such as primary care,[6 107] only three (5%) of the 57 included trials were conducted in a primary care setting. Power was limited to quantify the pooled effect of this stratum, and $g$ did not reach statistical significance. This means that there is an urgent need for RCTs evaluating internet-delivered transdiagnostic treatments based on a broad recruitment strategies in the routine clinical context. It is also important when planning for future RCTs to reduce currently high levels of missing data. More studies evaluating video-delivered interventions are needed. Adequately powered RCTs comparing internet-delivered transdiagnostic treatments with other bona-fide treatments for depression and anxiety are also needed.

## Conclusion

Internet-delivered transdiagnostic treatments suitable for both depression and anxiety appear to have moderate effects compared with rudimentary controls, and significant but small effects beyond those of attention/engagement controls. Evidence is lacking with regard to routine care, the video format, children and adolescents and older adults.

**Author affiliations**
¹Centre for Psychiatry Research, Department of Clinical Neuroscience, Karolinska Institutet, Stockholm, Sweden
²Liljeholmen University Primary Health Care Centre, Academic Primary Health Care Centre, Region Stockholm, Stockholm, Sweden
³Department of Psychology, Uppsala University, Uppsala, Sweden
⁴Division of Psychology, Department of Clinical Neuroscience, Karolinska Institutet, Stockholm, Sweden
⁵Gustavsberg University Primary Health Care Centre, Academic Primary Health Care Centre, Region Stockholm, Stockholm, Sweden
⁶Division of Family Medicine and Primary Care, Department of Neurobiology, Care Sciences and Society, Karolinska Institutet, Stockholm, Sweden

**Acknowledgements** We thank the Karolinska Institutet library service, and in particular Sabina Gillsund and Emma-Lotta Säätelä, for contributing to the systematic database search. We would also like to thank all study authors who responded to our queries and contributed with data on request.

**Contributors** KK and EA led the study design and data collection process, with input from all authors. KK and EA conducted the statistical analysis. KK wrote the initial draft in close collaboration with EA. All authors reviewed the manuscript for intellectual content, made contributions and approved the final version. KK and EA had full access to all data and took responsibility for the integrity of the data and the accuracy of the presented analysis. KK and EA are both guarantors of the study.

**Funding** This work was supported by the Swedish Research Council (EA; identifier: 2021-06496) and the Krica Foundation. Neither of the funders had any role in the design, execution, or publication process.

**Competing interests** KK, AHB, EHL, EL, JH, and EA have taken part in original research on internet-delivered treatment including transdiagnostic treatment.

**Patient and public involvement** Patients and/or the public were not involved in the design, or conduct, or reporting or dissemination plans of this research.

**Patient consent for publication** Not applicable.

**Ethics approval** Not applicable.

**Provenance and peer review** Not commissioned; externally peer reviewed.

**Data availability statement** Data are available upon reasonable request.

**ORCID iDs**
Karoline Kolaas http://orcid.org/0000-0002-5508-6919
Anne H Berman http://orcid.org/0000-0002-7709-0230
Erik Hedman-Lagerlöf http://orcid.org/0000-0002-7939-9848
Elin Lindsäter http://orcid.org/0000-0003-2547-9196
Jonna Hybelius http://orcid.org/0000-0002-1593-3431
Erland Axelsson http://orcid.org/0000-0003-2562-2925

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
