## [Reviewer comments · BMJ Open]

ARTICLE DETAILS

TITLE (PROVISIONAL)	Internet-delivered transdiagnostic psychological treatments for individuals with depression, anxiety, or both: a systematic review with meta-analysis of randomized controlled trials
AUTHORS	Kolaas, Karoline; Berman, Anne H.; Hedman-Lagerlöf, Erik; Lindsäter, Elin; Hybelius, Jonna; Axelsson, Erland

VERSION 1 – REVIEW

REVIEWER	Humpston, Clara University of York
REVIEW RETURNED	23-Jun-2023

GENERAL COMMENTS	I read this systematic review and meta-analysis with great interest. I find it generally well-written and provides new evidence about the effectiveness of internet-delivered psychological therapies for anxiety and depression. I have a few comments by page and line number below. Page 4, Line 26 (Abstract) - I am not sure if the unique contributions of the current study belong in the eligibility criteria section. Page 4, Line 38 - I am not entirely sure what is meant by 'controlled effect'. Did the authors mean random-effects models? I could be mistaken but this is not entirely clear or well-defined. Page 5, Line 17. What is meant by 'small-to-moderate varying'? I assume the authors meant that the effect sizes varied between small and moderate? Page 5, Line 55. I assume by 'treatment schools' the authors meant frameworks such as CBT or mindfulness (schools of thought). However, to avoid confusion may I suggest that the authors change this term to 'treatment frameworks'? Page 5, Line 57. I am not sure what 'reduction of range' means here. Page 6, Line 5 (Introduction) - it might be better to focus on depression and anxiety rather than starting with 'mental disorders....' which is far too broad in my view. Page 6, Line 12. I am not entirely convinced that most patients with depression and/or anxiety are untreated, surely this requires more context/nuance? The references put this in a global context (which may well be the case) however it needs to be made clearer (e.g., in developing countries)? Page 8, Line 47 (Methods). There is a difference between efficacy
--

	and effectiveness - and I am certain the authors speak of effectiveness and not efficacy of these interventions. Page 10, Line 27. 'Studies were required to lend themselves to the tabulation of means and standard deviations' sounds a little strange, may I suggest that the authors reword it to 'studies were required to have reported/calculated means and standard deviations'? Page 21, Line 22 (Discussion) - 'transdiagnostic treatments specifically for anxiety disorder' sounds quite self-contradictory? Surely if it is specific towards treating anxiety disorders it cannot be truly transdiagnostic. I hope the authors find my comments helpful in preparing a revision. Clara Humpston, University of York, UK
--	---

REVIEWER	Titov, Nick Macquarie University, School of Psychological Sciences
REVIEW RETURNED	30-Jun-2023

GENERAL COMMENTS	Thank you for inviting me to review this interesting and promising study. This is a well structured and organised manuscript on an interesting and important topic. I regret that my review has been somewhat curtailed by my uncertainty regarding the selection of included studies. I apologise in advance if I have misunderstood the Methods/Design, but I was surprised at the very low inclusion rate of RCTS (57 of 435 RCTs were included). I have reviewed the eligibility criteria and the supplementary materials but I remain unsure why a large number of RCTs, which appear to meet the inclusion criteria, have been omitted from this study (e.g., Berger et al. 2016; Dear et al. 2015; Fogliati et al. 2016). It would be helpful if the authors could clarify their description of the exclusion/inclusion criteria and include additional explanation of why so many studies were omitted and the possible impact on their results and conclusions. I would also encourage the authors to reconsider their comments in the Discussion about the absence of outcome data about transdiagnostic treatments in routine care. They may wish to refer to other studies that have explored these questions (e.g., Etzelmüller et al. 2020). I wish the authors well with their important work.
--

REVIEWER	Joffe, Grigori University of Helsinki, Psychiatry
REVIEW RETURNED	21-Jul-2023

GENERAL COMMENTS	The research explores a wide range of various internet-delivered therapies for depression or anxiety, or both. Given the ever-growing burden of these conditions for society; limited accessibility and affordability of face to face psychosocial treatments, as well as some
--

	advantages of transdiagnostic modes over the diagnosis. specific ones, the research topic is more than welcome. The paper is well written and easy to read. The applied methodology is solid, as is the statistical analysis. Some of the references are old, but in most cases the choice is appropriate due to the fundamental role of those earlier publications (although more recent could substitute for some of those, e.g., for the Wei et al., 2005). The paper is one of the rare ones where as a reviewer I do not see any need of additional elaboration, not to speak about any major concerns. The only minor ones are some inconsistencies in the use of "or/and" (for instance, in the eligibility criteria (c) and (d) there is "...treatment for anxiety and depression", while elsewhere it is "...depression, anxiety, or both...". So, finally: had the subjects to demonstrate both or each of the two would suffice?). Also a brief rationale for exclusion of "...studies that required all participants to meet full diagnostic criteria for both depression and an anxiety disorder at the same time" would be helpful for a reader. There are some typos, like "cut-off or symptoms" on p.9 (should it be "cut-off of symptoms"?), or "in in" (in Table 1). Many thanks for the opportunity to read this interesting and useful article.
--	--

REVIEWER	Afonso, José University of Porto, Faculty of Sport
REVIEW RETURNED	04-Oct-2023

GENERAL COMMENTS	Thank you for the opportunity to revise this manuscript. At the request of the editors, my focus was on methodology. I am not expert on the specific topic, and therefore my comments may be naïve. However, some conceptual aspects of the work directly influence the methodology of systematic reviews, and so I had to address them whenever necessary. Moreover, as an "outsider", perhaps my questions better reflect those of a non-expert reader, and that may help devising a manuscript with broader interest, i.e., targeting a wider readership. Title: It should read "systematic review with meta-analysis" (instead of "and meta-analysis"). Abstract:  • The first two statements of the abstracts seem to broad, and especially the second statement would benefit from a brief expansion of why online treatments show promise. I understand that writing an abstract is very difficult and that word constraints limit the authors' ability to provide more in-depth information. However, abstracts for systematic reviews are often allowed to be larger. Therefore, in case there is space, please develop the initial selling point a bit more. • The eligibility criteria should have been presented in the PICOS/PECOS order. More comments of these criteria in the review section devoted to methods. • Data extraction and synthesis: too unclear, in my opinion (as described in the abstract). Criteria for pooling data? What was the meta-analytical model used? Was the impact of heterogeneity assessed (e.g., I²)? Was risk of bias assessed? Was certainty of evidence (i.e., GRADE) assessed? Even being an abstract, more information should have been provided. • The abstract states that 16% of trials recruited exclusively from routine care; what about the other trials (the majority)? Then, it is stated that 5% delivered treatment via video. What about the
---

	majority (the missing 95%)?  • It is stated that for adults in treatment, large within-group reductions were seen in symptoms of depression and anxiety. But, since the goal was to assess RCTs, it is important to know: compared to what? For example, waitlist controls and treatment-as-usual should not be in the same bag. At a minimum, they should be targeted by subgroup analyses. How effective were the online interventions compared to waitlists, versus how effective they were compared to other interventions? And what exactly is “treatment-as-usual”? Were these well described and comparable across trials, both qualitatively (e.g., type of intervention) and quantitatively (e.g., weekly sessions)? And what does it mean to compare online interventions to attention/engagement controls? Can attention/engagement not be targeted by some online interventions as well? What specific online interventions worked the best? Are all online interventions the same? What was the compliance of the participants in interventions versus controls, and could this have explained part of the differences? I understand this is an abstract, so I’m merely highlighting doubts that will likely emerge on the readers. Unfortunately, the manuscript itself did little to answer these questions. • The authors state that heterogeneity was substantial, but no mention to heterogeneity had been mentioned in the methods section of the abstract. What was the specific statistic used? And what was the threshold for “substantial”? Were subgroup and/or sensitivity analyses performed to explore the sources of such heterogeneity, as recommended by PRISMA 2020 and by Cochrane guidelines? • How were adolescents defined? What were the age ranges that allowed classification as being adolescents? Also, what was the threshold used to define older adults? Since the threshold varies from 60 to 65 years, depending on the guidelines, what was the cut-off value established by the authors of this systematic review? • The conclusion of the abstract seems to bold in light of the results that were presented. The last sentence could be legitimate in the manuscript, but does not emerge naturally in the abstract, as it does follows from what was presented previously. Rationale  • Throughout the introduction, I failed to find a satisfying operational definition of “transdiagnostic treatments”. What does this term include and exclude, exactly? Without such an operational definition, serious concerns are raised concerning the study selection and, as such, concerning the representativeness of this review’s results. In fact, the definitions only appear late in the methods section. • The authors attempted to establish the relevance of the topic, but not the relevance of a systematic review. Are there others reviews on the topic (systematic or not)? What were their findings? Why is this particular review needed? • Hypothesizing that any intervention is superior to a waitlist is not a great hypothesis. How did the authors expect the interventions to compare against controls receiving alternative interventions? Methods  • Registration: proper registration in PROSPERO on April 16, 2021, i.e., before the initial database searches. • Eligibility criteria: these must be defined prior to the search strategy, as they influence the search strategy. And this is clearly established in PRISMA 2020, which the authors said to have followed. There are additional problems: (i) why restrict to English
--	---

	language? At least, provide a rationale for this option; (ii) PICOS/PECOS order is not followed, so again the authors deviated considerably from the PRISMA 2020; (iii) why were cluster randomized trials excluded? No rationale was provided for that; (iv) what is a “mix of participants”? What was the operational definition used to implement selection criteria?; (v) criterion (e) is too vaguely defined; (vi) minimum of participants on comparator groups? Or did the minimum of 10 only apply to the intervention? If so, why?; (vii) why was the cut-off value of post-treatment assessment no later than 3 months from treatment termination? What was the rationale for this cut-off value?; (viii) “studies were required to lend themselves to the tabulation of means and SDs” – what were the conversion methods applied, when necessary?; (ix) no criterion is presented for comparators. Currently, the eligibility criteria are misplaced, ill-defined, and do not allow independent replication of the study selection process. Moreover, the criteria clearly do not follow the PRISMA 2020 recommendations.  • Databases: The choice of databases seem solid and comprehensive. Extending Web of Science searches to all collections could have resulted in a few extra relevant manuscripts, but at the risk of vastly increasing the number of (non-relevant) records. PubMed could have been used instead of Medline (as PubMed includes, but is not limited to Medline), but Cochrane Library usually retrieves many results from PubMed. EMBASE, another relevant database, is included in Cochrane Library. Potentially, maybe Scopus could have been considered, but overall I believe the number (and relevance) of consulted databases is more than sufficient. • Search strategy: the search strategy presented in the online supplement tries to be multiple and specific, and risk having missed relevant key terms. A broader, cleaner search strategy should have been devised, with a clear general combination of code lines explained in the document. Also, it is not clear at all why only publication from 1995 onwards were considered. What is the rationale presented for this choice? • Additional searches: international guidelines recommend that, beyond database searches, additional steps are taken. These should include manual search in the reference lists and/or snowballing citation tracking of the included articles, as well as consultation of (at least two) external experts (whom, by definition, cannot integrate the authors’ list). For example, consultation of external experts is assessed by AMSTAR-2. These steps ensure a more comprehensive analysis of the relevant literature and may allow identification of manuscripts that were not indexed in the consulted databases. Why were these steps not taken? • Study selection methods: again, these should have been reported after the eligibility criteria, as per PRISMA 2020. • Data extraction: gender is complex construct – did the authors extract information on gender, or merely on sex? Why were important data extraction topics not considered in subgroup and/or sensitivity analyses (e.g., symptom severity, delivery format, etc.)? Why was a cut-off value established between interventions lasting <6 weeks versus ≥6 weeks – rationale for this specific value? Also, if waitlists and treatment-as-usual could not be properly differentiated, as the authors admit, why not simply merge them into a single category and stop mentioning both separately? • Risk of bias in individual studies: why was this information not used in the abstract to at least qualify the interpretation of the results? And why omit items related to blinding? RoB 2’s algorithm does not automatically raise risk of bias due to that. And, indeed, that item
--	--

should have been assessed, as it usually provides very relevant information. That item may even influence other issues, such as potential deviations from intended interventions. So, omission should not have been the implemented solution for items related to blinding. Finally, the decision to stop independent verification after 5 consecutive coincidental assessments is ill-advised; often, a few selected papers present special problems of analyses that may require multiple assessments and generate discussion between research team members.

- Statistical analysis: why was within-group meta-analysis the main goal? This seems incoherent with the criterion of the studies being RCTs. The main focus should have been the comparison with other interventions (or passive controls). Also, what specific random effects model was used? Please report. The I2 does not measure heterogeneity, but the impact of heterogeneity, as stated by the creator of the test himself.

- Publication bias: please replace with “risk of publication bias”, and refrain from interpreting the results as presence or absence of publication bias, as these tests have many other explanations for their results beyond publication bias. Also, a minimum of 10 studies per comparison should have been pre-determined, as no formal test should be applied with less than 10 studies. In fact, the existing tests would formally require well above 10 studies for achieving the needed statistical power, but the minimum of 10 is a usually accepted rule of thumb. However, the guidelines are clear: do not run those tests when <10 studies are available per comparison. Very important: per comparison.

- Why was certainty of evidence (again, included in PRISMA 2020) not assessed? Why was GRADE not applied?

Results

- Since there are problems with the eligibility criteria, the study selection process is not replicable by an independent team. Therefore, this casts a shadow over the results.

- What was the criteria for determining that studies would be pooled for age groups? For example, if an article presented an age range of 16 to 25, with only pooled data (i.e., not separated by age), in what category was this article classified? And, in such cases, were the authors of the original study contacted for further information? Similarly, what are “working-age adults”? And does this not vary by country? How was this defined?

- Were trials fully supervised? Partially supervised? Did they involve “homework”? Was compliance and adherence assessed? Did compliance and adherence differ between interventions and controls? The study characteristics lack highly relevant contextual information.

- Effects on depression and anxiety: what was the number of studies and the sample size for each particular comparison?

- Publication bias: was the minimum of 10 studies per comparison considered, or were the tests performed regardless? Also, remember: these tests are not indicative of publication bias or its absence. For a recent overview on the topic, please consult <https://link.springer.com/article/10.1007/s40279-023-01927-9>.

- GRADE is lacking to better frame the results.

Discussion

- Multiple statements in the discussion should be reframed considering the risk of bias assessments, the limitations of tests for assessing risk of publication bias, and including the much-needed GRADE results.

	 • Most topics of the discussion require a more in-depth discussion, including the role of potentially relevant intervention-related factors (e.g., compliance). • The limitations should clearly differentiate what are the limitations of the original studies, and the limitations of this review. Overall, I believe this to be a relevant work, but currently requiring much improvement.
--	---

REVIEWER	Yang, Jiang Henan University of Chinese Medicine
REVIEW RETURNED	06-Oct-2023

GENERAL COMMENTS	There is variability between studies, not only in the patient population in relation to the depression, anxiety disease-state, but also in how this is reported, i.e., the interventions. Accordingly, there is considerable study heterogeneity which would prevent pooling of data. Internet-delivered transdiagnostic psychological treatments is a mode of delivery of psychological treatments and therefore the specific aims and objectives, interventions prescribed, and therapeutic approaches, can differ. This current paper has not sufficiently handled these variations in the design. Consider citing the 57 included studies. Overall, the Eligibility criteria need to be further considered. The number of included studies is large, but the heterogeneity among the interventions is large, and the reliability of the results needs to be verified. It is recommended to narrow the research topic.
--

VERSION 1 – AUTHOR RESPONSE

Reviewer 1 general comment:

“I read this systematic review and meta-analysis with great interest. I find it generally well-written and provides new evidence about the effectiveness of internet-delivered psychological therapies for anxiety and depression. I have a few comments by page and line number below.”

We respond:

Thank you. We are pleased to hear that Reviewer 1 found the paper well written, and we agree that this work contributes with new evidence that could be valuable for the field.

Reviewer 1 comment #1:

“Page 4, Line 26 (Abstract) - I am not sure if the unique contributions of the current study belong in the eligibility criteria section.”

We respond:

Addressed. This section no longer mentions “unique contributions”. Now reads (p 3, lines 13-14): “This review concerned all treatment frameworks, both guided and unguided formats, and all age groups.”

Reviewer 1 comment #2:

“Page 4, Line 38 - I am not entirely sure what is meant by 'controlled effect'. Did the authors mean random-effects models? I could be mistaken but this is not entirely clear or well-defined.”

We respond:

We apologize for this not being entirely clear in the original manuscript. In our experience, a “controlled effect” is synonymous with a “between-group effect”, i.e., the difference in effect between an intervention and a control group. For clarity, we now use “between-group effect” instead of “controlled effect” throughout the manuscript, and the sentence referred to by Reviewer 1 now simply reads (p. 3, line 16-17): “We estimated pooled effects on depression symptoms and anxiety in terms of Hedges’ g with 95% CIs.”

Reviewer 1 comment #3:

“Page 5, Line 17. What is meant by 'small-to-moderate varying'? I assume the authors meant that the effect sizes varied between small and moderate?”

We respond:

Yes, we apologize for the missing comma. Now reads (p.4 line 7-8): “Internet-delivered transdiagnostic treatments for depression and anxiety show small to moderate added effects, varying by control condition.”

Reviewer 1 comment #4:

“Page 5, Line 55. I assume by 'treatment schools' the authors meant frameworks such as CBT or mindfulness (schools of thought). However, to avoid confusion may I suggest that the authors change this term to 'treatment frameworks'?”

We respond:

We thank Reviewer 1 for this suggestion. We have replaced the term “treatment schools” with “treatment frameworks” throughout the paper.

Reviewer 1 comment #5:

“Page 5, Line 57. I am not sure what 'reduction of range' means here.”

We respond:

We apologize if this was unclear. The original bullet point read: “Our analyses of moderators were based exclusively on trial-level characteristics. This implies poor power, and often restriction of range.” What we wanted to convey is simply that in this conventional type of meta-analysis where moderators are coded entirely on the study level (e.g., mean age, proportion of women, mean symptom severity) as opposed to the level of the individual patient (e.g., actual age, gender, or the symptom severity of an individual). A consequence of this is that certain moderator tests can be of limited value for those interested in effects on the level of the individual. Take age for example. In an individual clinical trial, it is not uncommon to recruit patients within a range of, say, 18 to 70. But it is unusual to see a clinical trial with a *mean* age of 18 or 70. Rather, means tend to cluster around 35-45 or so. In other words, on the level of studies, there is restriction of range in the age variable. Even if, hypothetically, age would have a moderating effect, we would have to aggregate a very large number of clinical trials with mostly minor differences in their mean age estimate to detect this. In other words, statistical power is limited. In the revised (R1) manuscript, we have rephrased this bullet point so that it simply reads (p. 5, line 1): “Our analyses of moderators were based exclusively on trial-level characteristics, and should preferably be replicated based on individual patient data.”

Reviewer 1 comment #6:

“Page 6, Line 5 (Introduction) - it might be better to focus on depression and anxiety rather than starting with 'mental disorders....' which is far too broad in my view.”

We respond:

Fair point. This sentence now reads (p 5, line 4): “Depression and the common anxiety disorders are associated with immense worldwide healthcare costs and disease burden.”

Reviewer 1 comment #7:

“Page 6, Line 12. I am not entirely convinced that most patients with depression and/or anxiety are untreated, surely this requires more context/nuance? The references put this in a global context (which may well be the case) however it needs to be made clearer (e.g., in developing countries)?”

We respond:

Having reconsidered this empirical statement, at least in the population as a whole, we believe there to be convincing evidence of a substantial and genuinely global “treatment gap” between the number of individuals meeting criteria for depression and the common anxiety disorders, and the number receiving treatment. For example, the study by Thornicroft et al. ¹ that we refer to with regard to depression surveyed 21 countries – including 6 categorized by the World Bank as belonging to the upper-middle income category and 5 the lower-middle income category – and found evidence of a considerable treatment gap in all countries. This gap was present even in high-income countries that usually score high in indicators of welfare such as the Human Development Index. One notable such country was Germany where 32% of those meeting full criteria for depression, and $0.79 \times 0.66 = 52\%$ of those deemed to be in clear need of treatment, received at least a minimally adequate treatment. Corresponding figures for the Netherlands were 33% and $0.82 \times 0.66 = 54\%$ ¹. Unsurprisingly, however, the treatment gap was most pronounced in the lower to middle income countries which included Colombia, Iraq, Nigeria (poor

data), Peru, and China (poor data). We find it unlikely that countries belonging to the lowest income category – mostly countries in sub-Saharan Africa such as the Central African Republic, the Democratic Republic of Congo, Liberia, Mali, Somalia, and Sudan – would have fared better, had such countries been included in the study.

We recognize that the sentence that Reviewer 1 is referring to concerns patients as opposed to the general population as a whole. We have therefore opted for a minor revision of this sentence, where we changed “most” to “many”, and “remain untreated” to “remain untreated or receive inadequate treatment”. The revised sentence thus reads (P 5, line 5-6): “However, many patients with depression and anxiety disorders remain untreated or receive inadequate treatment,^{1 2} illustrating the need for further development and dissemination of effective therapies”.

Reviewer 1 comment #8:

“Page 8, Line 47 (Methods). There is a difference between efficacy and effectiveness - and I am certain the authors speak of effectiveness and not efficacy of these interventions.”

We respond:

We thank Reviewer 1 for bringing this erratum to our attention, and apologize for the confusion. We certainly agree that there is a convention whereby “efficacy” is used to refer to the evaluation of effects in a highly controlled setting (typically a randomized controlled trial [RCT]), whereas “effectiveness” is used to refer to effects seen in the real-world routine care environment³. Within this framework, it should be uncontroversial to regard most RCTs included in this systematic review as having concerned efficacy rather than effectiveness, considering that few RCTs, 9/57 (16%) including student counselling, were conducted in a routine care setting. We have now revised the manuscript as a whole to ensure that, whenever appropriate, the language remains neutral as to whether effects are derived from efficacy or effectiveness trials. Consequently, the sentence referred to by Reviewer 1 now simply reads (p 8, line 11-12): “Studies were required to evaluate the effect of an internet-delivered transdiagnostic psychological treatment for individuals with anxiety, depression, or both.” The word “effectiveness” is

still used in the manuscript, but exclusively to refer to routine care findings which is standard practice³.

Reviewer 1 comment #9:

“Page 10, Line 27. 'Studies were required to lend themselves to the tabulation of means and standard deviations' sounds a little strange, may I suggest that the authors reword it to 'studies were required to have reported/calculated means and standard deviations'?”

We respond:

Addressed, now reads (P 10, line 8-13): “Studies were required to have reported means and standard deviations, or provided information that made it possible to derive such estimates. Typically, this was based on the original publication. In certain cases, conversions were applied, for example when the standard deviation could be derived from the sample size and the standard error of the mean. Whenever necessary, the author was also contacted and encouraged to provide estimates via email.”

Reviewer 1 comment #10:

“Page 21, Line 22 (Discussion) - 'transdiagnostic treatments specifically for anxiety disorder' sounds quite self-contradictory? Surely if it is specific towards treating anxiety disorders it cannot be truly transdiagnostic.”

We respond:

We agree and have now revised the sentence for increased clarity (P 22, line 4-6): “Especially for children and adolescents, we found this rather surprising considering the relatively long tradition of treatments intended to suit a wide spectrum of anxiety-, but not necessarily mood, disorders.”

Reviewer 2 general comment:

“Thank you for inviting me to review this interesting and promising study. This is a well structured and organised manuscript on an interesting and important topic. I regret that my review has been somewhat curtailed by my uncertainty regarding the selection of included studies.”

We respond:

We are pleased to hear that Reviewer 2 finds the manuscript well structured and organized, and also thank Reviewer 2 for acknowledging the importance of the topic. We appreciate the reviewer’s expertise in this area, regret that there was room for uncertainty, and have made a series of clarifications with regard to the eligibility criteria as is detailed below.

Reviewer 2 comment #1:

“I apologise in advance if I have misunderstood the Methods/Design, but I was surprised at the very low inclusion rate of RCTS (57 of 435 RCTs were included). I have reviewed the eligibility criteria and the supplementary materials but I remain unsure why a large number of RCTs, which appear to meet the inclusion criteria, have been omitted from this study (e.g., Berger et al. 2016; Dear et al. 2015; Fogliati et al. 2016). It would be helpful if the authors could clarify their description of the exclusion/inclusion criteria and include additional explanation of why so many studies were omitted and the possible impact on their results and conclusions.”

We respond:

We apologize that the inclusion and exclusion criteria were not described clearly enough in the original manuscript. All three studies cited by Reviewer 2 above were found in the literature search, and were subsequently excluded on the following criterion: “(d) Studies needed to recruit participants suffering from clinically significant depression, anxiety, or both. [...]”:

- Berger et al. ⁴ (Published online 2016). This study employed the following inclusion criterion “(2) the MD confirmed a SAD, PDA and/or GAD diagnosis”, meaning that a patient with primary depression and no anxiety diagnosis would not have been included.

- Dear et al. ⁵ This study employed the following inclusion criterion: *"(ii) a principal complaint of GAD symptoms"*, which would have excluded participants with primary depression without principal complaints of GAD symptoms.
- Fogliati et al. ⁶ This study employed the following inclusion criterion: *"(ii) principal symptoms consistent with Panic Disorder"*, which implies that patients with pure depression were not eligible.

To prevent future confusion with regard to the eligibility criteria, we have made clarifications throughout the R1 version of the manuscript. Criterion d now reads as follows (P 8-9): "Studies needed to recruit participants suffering from clinically significant depression, anxiety, or both. We considered this criterion to be met if one of the following was true: (i) all participants were required to meet full diagnostic criteria for either primary depression or a primary anxiety disorder and both populations were included in the trial, or (ii) all participants were required to score above a valid cut-off on an accepted screening measure for either depression or anxiety and both populations were included in the trial, or (iii) all participants were required to score above both a valid cut-off for depression and a valid cut-off for anxiety, or (iv) the baseline sample mean of at least one transdiagnostic treatment arm and at least one other arm were above both a valid cut-off for depression and a valid cut-off for anxiety. Studies that subsumed obsessive-compulsive disorder or post-traumatic stress disorder under the umbrella of anxiety disorders, in accordance with earlier DSM versions, were included. Studies that included a substantial proportion of subclinical participants were included only if clinical participants were reported separately."

Reviewer 2 comment #2:

"I would also encourage the authors to reconsider their comments in the Discussion about the absence of outcome data about transdiagnostic treatments in routine care. They may wish to refer to other studies that have explored these questions (e.g., Etzelmueller et al. 2020). I wish the authors well with their important work."

We respond:

We appreciate this comment about previous studies exploring acceptability and pre-post within-group improvement in the routine care setting, and have now clarified in the introduction that (P 6, line 5-9): “Fourth, although acceptability and within-group change has been studied to some extent in the routine care environment , based on randomized controlled trials that include a combination of patients with anxiety and depression, it is not known how well the effects of internet-delivered formats translate to routine clinical settings such as primary care”. As alluded to in our response to Reviewer 2 comment #1, it appears to us that the systematic review by Etzelmueller et al also included RCTs where all participants were required to meet full criteria for one specific anxiety or mood disorder (e.g., all participants were required to meet full criteria for generalized anxiety disorder). Based on our review where we made an a priori decision to exclude such studies (because we wanted the recruitment strategy to be broader and thereby mirror a truly transdiagnostic approach), ultimately, only 3 RCTs of transdiagnostic Internet-delivered psychological treatments were conducted in primary care. We have now clarified in the discussion that when we speak of a shortage, we refer primarily to RCTs of transdiagnostic treatments where recruitment was broad (P 24, line 4-6): “This means that there is an urgent need for RCTs evaluating internet-delivered transdiagnostic treatments based on a broad recruitment strategies in the routine clinical context”. While we recognize that a case could be made for including studies such as that by Dear et al., we hope that Reviewer 2 can also see the value in complementing previous systematic reviews with the present one which had more of an emphasis on the recruitment strategy being broad in the sense of not selecting all study participants on one and the same particular psychiatric diagnosis.

Reviewer 3 general comment:

“The research explores a wide range of various internet-delivered therapies for depression or anxiety, or both. Given the ever-growing burden of these conditions for society; limited accessibility and affordability of face to face psychosocial treatments, as well as some advantages of transdiagnostic modes over the diagnosis specific ones, the research topic is more than welcome. The paper is well written and easy to read. The applied methodology is solid, as is the

statistical analysis. Some of the references are old, but in most cases the choice is appropriate due to the fundamental role of those earlier publications (although more recent could substitute for some of those, e.g., for the Wei et al., 2005)."

We respond:

We are delighted to hear that Reviewer 3 acknowledges the impact these conditions have on society, and welcomes research on this topic. We are also happy to hear that Reviewer 3 finds the paper well written and easy to read, and regards the methods and statistical analysis to have been robust. In line with the suggestion of Reviewer 3, we have replaced ⁷ with more recent publications (P 22, line 5): ⁸ and ⁹.

Reviewer 3 comment #1 (only specific comment):

"The paper is one of the rare ones where as a reviewer I do not see any need of additional elaboration, not to speak about any major concerns. The only minor ones are some inconsistencies in the use of "or/and" (for instance, in the eligibility criteria (c) and (d) there is "...treatment for anxiety and depression" , while elsewhere it is "... depression, anxiety, or both...". So, finally: had the subjects to demonstrate both or each of the two would suffice?).

Also a brief rationale for exclusion of "...studies that required all participants to meet full diagnostic criteria for both depression and an anxiety disorder at the same time" would be helpful for a reader. There are some typos, like "cut-off or symptoms" on p.9 (should it be "cut-off of symptoms"?) or "in in" (in Table 1). Many thanks for the opportunity to read this interesting and useful article."

We respond:

Again, we are happy to hear that Reviewer 3 views the manuscript as interesting and suitable for publication. We also appreciate these minor suggestions for improved readability. As to the first question, the text (p 7, line 3 and p 8, line 24-25) now reads: "depression, anxiety, or both". As has now been clarified (p 8-9), "[w]e considered this criterion to be met if one of the following was true: (i) all participants were required to meet full diagnostic criteria for either primary depression or a primary anxiety disorder and both populations were included in the trial, or (ii) all participants wererequired to

score above a valid cut-off on an accepted screening measure for either depression or anxiety and both populations were included in the trial, or (iii) all participants were required to score above both a valid cut-off for depression and a valid cut-off for anxiety, or (iv) the baseline sample mean of at least one transdiagnostic treatment arm and at least one other arm were above both a valid cut-off for depression and a valid cut-off for anxiety.” Please also see Reviewer 2, comment #2.

In line with the suggestion of Reviewer 3, we have also added a brief rationale for the exclusion of “studies that required all participants to meet full diagnostic criteria for both depression and an anxiety disorder at the same time”. This reads (P 9, line 8-17): “We excluded RCTs that either required all participants to meet full formal diagnostic criteria for depression, or required all participants to meet full formal diagnostic criteria for an anxiety disorder. This was to ensure a reasonably broad recruitment strategy, and to further reduce the risk that the treatment had focused overwhelmingly on one specific psychiatric disorder. We also excluded RCTs that required all participants to meet full formal diagnostic criteria for both depression and an anxiety disorder at the same time. This is a very unusual design that requires all participants to have a very substantial level of symptomatology that we suspected could be indicative of a highly specialized care setting unlike primary care which we were primarily interested in.” We have also corrected the last minor typos, and, again, thank Reviewer 3 for the generally positive feedback.

Reviewer 4 general comment:

“Thank you for the opportunity to revise this manuscript. At the request of the editors, my focus was on methodology. I am not expert on the specific topic, and therefore my comments may be naïve. However, some conceptual aspects of the work directly influence the methodology of systematic reviews, and so I had to address them whenever necessary. Moreover, as an “outsider”, perhaps my questions better reflect those of a non-expert reader, and that may help devising a manuscript with broader interest, i.e., targeting a wider readership.”

We respond:

We thank Reviewer 4 for this ambitious review of our manuscript, with particular emphasis on methodological considerations. We recognize that this was an unusually detailed reading,

and appreciate that Reviewer 4 shared his professional views on how to further improve the manuscript. We have gone through the comments carefully, and made systematic attempts to improve the text further. We wish to emphasize, however, that this was not always possible while adhering to the journal guidelines (e.g., the 300 word limit for the abstract).

Reviewer 4 comment #0:

“Title: It should read “systematic review with meta-analysis” (instead of “and meta-analysis”).”

We respond:

The title now reads: “Internet-delivered transdiagnostic psychological treatments for individuals with depression, anxiety, or both: a systematic review with meta-analysis of randomized controlled trials”.

Reviewer 4 comment #1a (“Abstract”):

“• The first two statements of the abstracts seem to broad, and especially the second statement would benefit from a brief expansion of why online treatments show promise. I understand that writing an abstract is very difficult and that word constraints limit the authors’ ability to provide more in-depth information. However, abstracts for systematic reviews are often allowed to be larger. Therefore, in case there is space, please develop the initial selling point a bit more.

• Data extraction and synthesis: too unclear, in my opinion (as described in the abstract). Criteria for pooling data? What was the meta-analytical model used? Was the impact of heterogeneity assessed (e.g., I²)? Was risk of bias assessed? Was certainty of evidence (i.e., GRADE) assessed? Even being an abstract, more information should have been provided.

• The abstract states that 16% of trials recruited exclusively from routine care; what about the other trials (the majority)? Then, it is stated that 5% delivered treatment via video. What about the majority (the missing 95%)?

• The authors state that heterogeneity was substantial, but no mention to heterogeneity had been mentioned in the methods section of the abstract. What was the specific statistic used? And

what was the threshold for “substantial”? Were subgroup and/or sensitivity analyses performed to explore the sources of such heterogeneity, as recommended by PRISMA 2020 and by Cochrane guidelines?

• How were adolescents defined? What were the age ranges that allowed classification as being adolescents? Also, what was the threshold used to define older adults? Since the threshold varies from 60 to 65 years, depending on the guidelines, what was the cut-off value established by the authors of this systematic review?”

We respond:

We thank Reviewer 4 for this input and regard much of the feedback given here in comment #1a to be sound and reasonable. We agree that abstracts for systematic reviews are often allowed to be longer, not least to accommodate the PRISMA 2020 items. Here, we were required to stay within 300 words and to follow the journal guidelines of BMJ

Open (<https://bmjopen.bmj.com/pages/authors>). Unfortunately, this does not make it possible to make comprehensive changes or to implement most of the suggestions above. If the Editor so wishes, we would of course be willing to extend the abstract in order to provide more detailed information to accommodate more of the suggestions by Reviewer 4. Within the framework of 300 words and the journal guidelines, we have been able to make the following changes:

- In order to stay within the word limit while also avoiding vagueness, we have simply deleted the sentence “Online psychological treatments show promise.”
- We have clarified that effects were pooled using “random effects meta-analysis”, and that “[a]bsolute and relative heterogeneity was quantified as the τ^2 and I^2 ”.
- We have added the GRADE rating, i.e., that “the certainty of the evidence was very low”. See also Reviewer 4, comment #13.

With regard to the age criteria, this has been clarified in the main text. See Reviewer 4, comment #15.

Reviewer 4 comment #1b (“Abstract”):

“• The eligibility criteria should have been presented in the PICOS/PECOS order. More comments of these criteria in the review section devoted to methods.

• It is stated that for adults in treatment, large within-group reductions were seen in symptoms of depression and anxiety. But, since the goal was to assess RCTs, it is important to know: compared to what? For example, waitlist controls and treatment-as-usual should not be in the same bag. At a minimum, they should be targeted by subgroup analyses. How effective were the online interventions compared to waitlists, versus how effective they were compared to other interventions? And what exactly is “treatment-as-usual”? Were these well described and comparable across trials, both qualitatively (e.g., type of intervention) and quantitatively (e.g., weekly sessions)? And what does it mean to compare online interventions to attention/engagement controls? Can attention/engagement not be targeted by some online interventions as well? What specific online interventions worked the best? Are all online interventions the same? What was the compliance of the participants in interventions versus controls, and could this have explained part of the differences? I understand this is an abstract, so I’m merely highlighting doubts that will likely emerge on the readers. Unfortunately, the manuscript itself did little to answer these questions.

• The conclusion of the abstract seems to bold in light of the results that were presented. The last sentence could be legitimate in the manuscript, but does not emerge naturally in the abstract, as it does follows from what was presented previously.”

We respond:

Our impression is that the PRISMA 2020 does not mention the PICO framework, nor does it appear to prescribe that the elements of the PICO should be presented in a particular order. The introductory paper states explicitly that although the “*PRISMA 2020 provides a template for where information might be located, the suggested location should not be seen as prescriptive; the guiding principle is to ensure the information is reported*”¹⁰. We note that an adapted version of the PRISMA has been published in the field of sports medicine and musculoskeletal rehabilitation,¹⁰ 11 and that this version, unlike the conventional PRISMA, does mention the PICO framework. Could it be that Reviewer 4 had this adapted version in mind? We followed the conventional

PRISMA 2020 which does not refer to the term “PICO” or prescribe an obligatory order of presentation.^{10 11} The abstract contains conventional information about the population, intervention, comparator, and outcome. We agree that the abstract is unusually brief for a systematic review, but as mentioned above this is due to the word limit.

With regard to the longer comment about control groups, we did present a planned series of moderator analyses (Table 1 in the manuscript) and wholeheartedly agree with Reviewer 4 that the term “treatment as usual” is broad. The latter is a recurrent critique within the field of psychotherapy that has a relatively long history, and that is debated from time to time (see for example: ¹²). Precisely for this reason, in this systematic review, we went to great lengths in ensuring that we did not classify all control groups referred to by the original authors as “treatment as usual” under the same category “at face value” (a procedure that is seemingly common in the field of psychotherapy, including in this journal, see for example: ^{13 14}). It is increasingly recognized that what is referred to as “treatment as usual” in original studies can either (a) be a very rudimentary control that is very similar to, and oftentimes even indistinguishable from, a waitlist control – or (b) be more reminiscent of a “psychological placebo” or credible intervention of some sort that yet does not meet the requirement of a bona-fide psychological treatment.¹⁵ It has been recommended to take this heterogeneity in account when conducting meta-analysis.¹⁶ We tried to follow this recommendation, and have therefore, regardless of what the original authors called their control groups, made a distinction between control groups that were very rudimentary, with no or almost no intervention (now referred to as “rudimentary controls”) and control groups that included some standardized procedure that was intended to control for the attention from a caregiver and the effect of taking part in an intervention (referred to as “attention/engagement controls”). We believe that this change of terminology has resulted in in greater clarity and an unusually meaningful analytic approach compared for example to the aforementioned examples. In summary, we have done more distinctions than is usually the case in this field.

As to the question of within-group effects, we also agree with Reviewer 4. There is really no controversy in that such effects do not necessarily reflect causality. Including such effects (mean change within the group receiving psychotherapy), is however common in systematic reviews, as can be seen for example in one of the few previous systematic reviews on broadly the same topic

as discussed here.¹⁷ This is primarily motivated by the need to assure that therapies have been delivered with maintained quality, that the lack of controlled effects do not merely reflect the lack of change overall, and also to provide crude reference estimates that can be used in contextualising naturalistic single-group prospective cohort studies in the routine care environment (consider for example the reference recommended in Reviewer 2's comment #2). Also, as suggested by Reviewer 4, it is important to consider sources for heterogeneity pertaining to the pooled between-group effects. The heterogeneity estimates that pertain to the within-group effects can give an indication as to whether a difference in the (main) active group within-group effect could drive differences in between-group effects. Again, we wish to emphasise that we believe that the average reader is aware that within-group effects reflect change within a group, and we see no problem with including within-group effects here as long as these are reported in an clear and transparent manner.

Last, while we respect the opinion of Reviewer 4, we disagree that the (now slightly updated) sentence "Research is needed regarding routine care, the video format, children and adolescents, and older adults" does not follow naturally from the preceding text of the abstract. In the results section of the abstract, we report the exact number of RCTs that were included with these characteristics which, in our opinion, makes the last sentence logical.

Reviewer 4 comment #2 ("Rationale"):

“• Throughout the introduction, I failed to find a satisfying operational definition of “transdiagnostic treatments”. What does this term include and exclude, exactly? Without such an operational definition, serious concerns are raised concerning the study selection and, as such, concerning the representativeness of this review’s results. In fact, the definitions only appear late in the methods section.

• The authors attempted to establish the relevance of the topic, but not the relevance of a systematic review. Are there others reviews on the topic (systematic or not)? What were their findings? Why is this particular review needed?

• *Hypothesizing that any intervention is superior to a waitlist is not a great hypothesis. How did the authors expect the interventions to compare against controls receiving alternative interventions?"*

We respond:

The lack of a formal definition of "transdiagnostic treatments" is an excellent point. We thank Reviewer 4 for pointing this out, apologize for the inconvenience, and have clarified on p 5, line 11-14 (R1 version) that the term refers to: "psychological treatments designed to suit patients suffering from a range of different types of depression and anxiety problems. That is, transdiagnostic treatments are characterized by focusing on components relevant for both depression and anxiety, in contrast to diagnosis-specific treatments targeting one specific mental health condition."

Regarding previous meta-analyses, we cite such examples in several segments of the manuscript and have now clarified that certain empirical statements were based on meta-analyses specifically. Most notably (p. 5, line 17): "In early meta-analysis, transdiagnostic psychological treatments have been found to have moderate pooled effects on depression and anxiety compared to mostly rudimentary treatment-as-usual and waitlist control groups.¹⁸" And (p. 5-6, line 4): "For internet-delivered transdiagnostic psychological treatments, large within-group symptom reductions have been observed, with early meta-analyses reporting mixed outcomes compared to a heterogeneous range of control groups, and comparisons versus diagnosis-specific treatments typically resulting in relatively similar effects on depression and general anxiety.^{17 19 20}"

We agree with Reviewer 4 that the waitlist control condition is weak, but including comparisons with rudimentary controls is standard practice because this helps most readers to put the effects in a larger context of psychological treatment, and facilitates comparison with previous reviews. As is reported in the main manuscript, we also include other comparators in separate analyses to the extent that this was possible. Prior to conducting this review, we hypothesized that controlled effects on anxiety and depression versus disorder-specific treatments would be small (0.3 standardized units or less). However, we did not stipulate any a priori hypothesis with regard to the comparison to attention/engagement controls, as disorder-specific treatments were considered the most important. As it turned out, there were only

three comparators that could be regarded as bona-fide treatments. Because we had decided a priori only to conduct meta-analyses when we had at least four comparison, we conducted no meta-analysis focusing on bona-fide controls. We see no reason to comment further on this in the manuscript.

Reviewer 4 comment #3 (“Methods”):

“Registration: proper registration in PROSPERO on April 16, 2021, i.e., before the initial database searches.”

We respond:

Now explicitly reads (p. 7, line 6): “This systematic review with meta-analysis adhered to the PRISMA 2020 statement¹⁰ and was registered at PROSPERO (CRD42021243172) on April 16, 2021 and at the Open Science Framework (<https://osf.io/dtcey>) on May 18, 2021, i.e., before the initial database searches.”

Reviewer 4 comment #4a (“Methods”):

“Eligibility criteria: these must be defined prior to the search strategy, as they influence the search strategy. And this is clearly established in PRISMA 2020, which the authors said to have followed. There are additional problems: (i) why restrict to English language? At least, provide a rationale for this option; (ii) PICOS/PECOS order is not followed, so again the authors deviated considerably from the PRISMA 2020; (iii) why were cluster randomized trials excluded? No rationale was provided for that”

We respond:

Our impression is that the PRISMA does not prescribe that the eligibility criteria must be defined prior to the search strategy. Rather, the PRISMA 2020 explanation and elaboration states that although the “PRISMA 2020 provides a template for where information might be located, the suggested location should not be seen as prescriptive; the guiding principle is to ensure the information is reported”²¹. In this case, we found it more reader-friendly to give some overarching

information first, before going into the eligibility criteria. If deemed necessary, we are of course prepared to change this order on the discretion of the Editor.

(i) We restricted this systematic review to articles written in English because the authors are able to read academic texts in English and Swedish, and Swedish peer-reviewed journal articles are extremely unusual in the field of psychology. To restrict the search to English is standard practice in this research field,^{17 22 23} and makes it possible for most international colleagues to review methodological decisions. We now acknowledge among the limitations that (p. 24, line 9-10, R1 version): "This review included studies published in English only, which means that RCTs published solely in other languages were systematically overlooked."

(ii) As we pointed out also in response to Reviewer 4 comment #1b, we do not interpret the PRISMA 2020 as recommending an obligatory order in which it is necessary to report the elements of the PICO.

(iii) Excellent point. We have now clarified that we decided to exclude cluster randomized trials primarily (p. 8, R1 version): "because our focus was on conducting meta-analyses, for which cluster randomized trials require distinct analytical methods."²⁴

Reviewer 4 comment #4b ("Methods"):

"(iv) what is a "mix of participants"? What was the operational definition used to implement selection criteria?; (v) criterion (e) is too vaguely defined; (vi) minimum of participants on comparator groups? Or did the minimum of 10 only apply to the intervention? If so, why?"

We respond:

(iv) We have now clarified that by a "mix of participants" we meant that (p. 9, line 3): "Studies needed to recruit participants suffering from clinically significant depression, anxiety, or both. We considered this criterion to be met if one of the following was true: (i) all participants were required to meet full diagnostic criteria for either primary depression or a primary anxiety disorder and both populations were included in the trial, or (ii) all participants were required to score above a valid cut-off on an accepted screening measure for either depression or anxiety and both populations were included in the trial, or (iii) all participants were required to score above both a valid cut-off for

depression and a valid cut-off for anxiety, or (iv) the baseline sample mean of at least one transdiagnostic treatment arm and at least one other arm were above both a valid cut-off for depression and a valid cut-off for anxiety.“

(v) After clarification, criterion e now reads (p. 9, line 1-5): “Studies were required to evaluate treatments for samples that were representative for a general medical or mental health clinical setting in the sense that the sample had not been heavily selected on any particular clinical characteristic or medical condition other than that of primary interest. This led to the exclusion for example of treatments specifically aimed at cancer patients with depression or anxiety.“

(vi) Regarding the criterion of 10 participants in the intervention arm this was chosen to provide a common sense minimum threshold so that we would not be required to include case series or the like, and was based on the assumption that in an RCT, all groups tend to be more or less similar in size. This is also how things turned out, so the fact that the criterion focused on the intervention group only (as opposed to a requirement of 10 + 10) had no practical implication for which studies that were included, or how these were analyzed.

Reviewer 4 comment #4c (“Methods”):

“(vii) why was the cut-off value of post-treatment assessment no later than 3 months from treatment termination? What was the rationale for this cut-off value?; (viii) “studies were required to lend themselves to the tabulation of means and SDs” – what were the conversion methods applied, when necessary?; (ix) no criterion is presented for comparators. Currently, the eligibility criteria are misplaced, ill-defined, and do not allow independent replication of the study selection process. Moreover, the criteria clearly do not follow the PRISMA 2020 recommendations.”

We respond:

(vii) This is a field-specific standard. In trials of Internet-based psychological treatments, it is common practice to conduct follow-up measurements 6 and 12 months after treatment termination. Though there are of course exceptions, we found it reasonable to require that all pooled outcomes should be collected within three months from treatment termination so that it would lie closer to a typical post-treatment assessment than a typical follow-up assessment. For some context and

introduction, see for example: ²⁵⁻²⁷. We have clarified that (p. 10): “Post-treatment assessment needed to be completed within three months after treatment termination, so as to not lie closer to a typical follow-up assessment.”

(viii) When information was missing to tabulate the outcomes we e-mailed the corresponding author. (For example of such articles see: ²⁸⁻³⁰). We have clarified this in the text that now reads (p. 10, line 13): “Studies were required to have reported means and standard deviations, or provided information that made it possible to derive such estimates. Typically, this was based on the original publication. In certain cases, conversions were applied, for example when the standard deviation could be derived from the sample size and the standard error of the mean. Whenever necessary, the author was also contacted and encouraged to provide estimates via email.”

(ix) The reason why no criterion was listed for the comparators is that, as is now clearly stated on p. 10 (R1 version): “[a]ll control groups were included”. We have also clarified that (p. 11): “In line with recent findings demonstrating that many treatment-as-usual control groups are virtually indistinguishable from waitlist controls,¹⁶ we did not lump all treatment-as-usual controls into the same category but rather made a distinction between (a) rudimentary controls that comprised waitlist controls and other controls without a structured intervention, (b) attention/engagement controls where the participants received some sort of standardized intervention that controlled for the attention from a caregiver or engagement in a study but without this being a conventional psychological treatment, and (c) bona-fide treatments “delivered by trained therapists and [...] based on psychological principles, [...] offered to the psychotherapy community as viable treatments.¹⁵” Having reviewed items i-ix, we hope that Reviewer 4 regards these as viable clarifications and improvements.

Reviewer 4 comment #5 (“Methods”):

“Databases: The choice of databases seem solid and comprehensive. Extending Web of Science searches to all collections could have resulted in a few extra relevant manuscripts, but at the risk of vastly increasing the number of (non-relevant) records. PubMed could have been used instead of Medline (as PubMed includes, but is not limited to Medline), but Cochrane Library usually retrieves many results from PubMed. EMBASE, another relevant database, is included in Cochrane Library.

Potentially, maybe Scopus could have been considered, but overall I believe the number (and relevance) of consulted databases is more than sufficient.”

We respond:

We appreciate that Reviewer 4 finds the number and relevance of consulted databases to be more than sufficient. This was by far the largest systematic review of internet-delivered transdiagnostic psychological treatments for depression and anxiety to date.

Reviewer 4 comment #6 (“Methods”):

“Search strategy: the search strategy presented in the online supplement tries to be multiple and specific, and risk having missed relevant key terms. A broader, cleaner search strategy should have been devised, with a clear general combination of code lines explained in the document. Also, it is not clear at all why only publication from 1995 onwards were considered. What is the rationale presented for this choice?”

We respond:

In the context of the current literature, we are rather of the opinion that the broad search strategy was an important strength of the present systematic review. We went to great lengths in ensuring that this was a broad and relevant search, and consulted the Karolinska Institutet library service which has extensive experience of developing similar literature searches for systematic reviews in the medical sciences. Our resulting literature search was more comprehensive than those conducted previously in the field, and we wish to emphasise that this was not a negligible improvement. See for example: ¹⁷ ¹⁸. The most recent meta-analysis of transdiagnostic treatments for depression and anxiety that was published this year was published in a relatively high-impact journal (Psychological Medicine). Even though that systematic review not only included digital but also face-to-face transdiagnostic treatments, ¹⁸ their search resulted in 45 included articles. This is to be compared with 57 included in our search, where, importantly, we focused on internet-delivered therapies specifically. The rationale for limiting search hits from 1995 onward is that (now p.

7) “there were no RCTs of Internet-delivered transdiagnostic psychological treatments before then”. This is a small enough field where we can say this with reasonable confidence.

Reviewer 4 comment #7 (“Methods”):

“Additional searches: international guidelines recommend that, beyond database searches, additional steps are taken. These should include manual search in the reference lists and/or snowballing citation tracking of the included articles, as well as consultation of (at least two) external experts (whom, by definition, cannot integrate the authors’ list). For example, consultation of external experts is assessed by AMSTAR-2. These steps ensure a more comprehensive analysis of the relevant literature and may allow identification of manuscripts that were not indexed in the consulted databases. Why were these steps not taken?”

We respond:

We thank Reviewer 4 for pointing this out. We did in fact manually review the reference lists of the two previous meta-analyses that we found most relevant.^{17 19} This should have been mentioned in the original manuscript and is now stated on p. 7 (R1 version): “We also reviewed the reference lists of previous meta-analyses that we found relevant.^{17 19}” Furthermore, we have now clarified that (p. 7): “Candidate search strings were validated against a selection of articles that we were aware of prior to this systematic review, and after discussion within the research group, as well as with an external expert in psychodynamic online therapies.” This was to ensure that this perspective was represented, as no one in the research group had expert knowledge concerning such therapies. Again, we wish to emphasise that this was by far the most comprehensive systematic review of internet-delivered transdiagnostic psychological treatments for depression and anxiety to this date.

Reviewer 4 comment #8 (“Methods”):

“Study selection methods: again, these should have been reported after the eligibility criteria, as per PRISMA 2020.”

We respond:

We do not interpret the PRISMA 2020 as prescribing any particular order in which all items need be reported.

Reviewer 4 comment #9 (“Methods”):

“Data extraction: gender is complex construct – did the authors extract information on gender, or merely on sex? Why were important data extraction topics not considered in subgroup and/or sensitivity analyses (e.g., symptom severity, delivery format, etc.)? Why was a cut-off value established between interventions lasting <6 weeks versus ≥6 weeks – rationale for this specific value? Also, if waitlists and treatment-as-usual could not be properly differentiated, as the authors admit, why not simply merge them into a single category and stop mentioning both separately?”

We respond:

In the medical sciences, there is a relatively long tradition of using “gender” to denote “a person’s self-representation as man or woman, or how that person is responded to by social institutions based on the individual’s gender presentation.”³¹ The alternative term, “sex”, is more commonly used to denote “the classification of living things as man or woman according to their reproductive organs and functions assigned by chromosomal complement”³¹ Based on this taxonomy, which we believe to still represent the mainstream of the field, we believe that it would be most logical to use the term “gender” as opposed to “sex” in the present work because the variable in question is almost universally based on self-report data, and more or less never based on the study of reproductive organs or chromosomes. We have therefore retained the term “gender”.

We were surprised by Reviewer 4’s comment that variables such as symptom severity and delivery format were supposedly not considered in subgroup or sensitivity analyses. In line with the preregistered protocol, we did conduct such moderator and subgroup analyses which are reported in a relatively extensive manner, primarily in Table 1. The cut-off value of <6 weeks versus ≥6 weeks was based on a lengthy discussion within the research group, and clinical judgement of what would typically be considered a brief vs. a more duration of psychotherapy. We appreciate the reviewer’s comment as in most cases we are against the dichotomization of continuous or count variables, primarily due to the loss of statistical power. In this particular case, however, we genuinely believed that it would be of more substantial clinical value to contrast what are usually considered brief therapies with therapies of a more

conventional length. We have now clarified on p. 11 (R1 version) that the dichotomous coding was “intended to contrast what are usually considered brief therapies with therapies of a more conventional length”

VERSION 2 – REVIEW

REVIEWER	Humpston, Clara University of York
REVIEW RETURNED	13-Dec-2023

GENERAL COMMENTS	I would like to thank the authors for addressing my previous comments in such detail. I have no further suggestions as I am happy with the changes/responses made.
--

REVIEWER	Titov, Nick Macquarie University, School of Psychological Sciences
REVIEW RETURNED	18-Dec-2023

GENERAL COMMENTS	Best of luck with your work.
------------------------------

REVIEWER	Afonso, José University of Porto, Faculty of Sport
REVIEW RETURNED	30-Dec-2023

GENERAL COMMENTS	Thank you for the efforts put in place to improve the manuscript and to clarify several aspects of the reporting of information. I am satisfied with the provided responses.
--